# The role of transthalamic pathways in perception
Kevin P. Koster[1] & Christina Mo [2] ✉

Cognition is supported by neuronal signaling between regions of the cerebral cortex, most of which are linked by a strong synapse in the thalamus, forming transthalamic pathways. These pathways have gained attention for their powerful influence on perception, distinct from direct corticocortical pathways, prompting a reassessment of current cortical processing models. Recent advances in recording and manipulation technologies have allowed components of these pathways to be probed during behavior, but not the entire pathway. Here we synthesize findings on transthalamic contributions to perceptual behavior and outline the methodological constraints that shape interpretations. We argue that, despite these limitations, a converging conceptual update is taking form: transthalamic pathways operate as dynamic integrators that convey contextual, internal-state, and task-relevant information across distributed cortical areas. More complete understanding of these circuits will refine broader theories of brain computation.

How the brain integrates what we sense from the outside world with our internal states to guide behavior remains one of the central puzzles in neuroscience. Many decades of research have revealed that different regions of the cerebral cortex make distinct contributions to this process, and silencing or recording from them during perception and decision-making has helped map out a functional hierarchy from sensory to association to motor areas[1–13]. Within this framework, cortical information flow is often discussed as depending primarily on direct connections between cortical regions[14,15]. However, this cortico-centric view is now being reappraised, as advances in circuit-dissection methods reveal the importance of intermediary routes. Silencing a cortical region disrupts not only its direct corticocortical projections (area A → area B), but also indirect communication pathways that involve additional brain structures.

Chief among these is the thalamus, the intricately connected partner of the cortex long cast as a passive relay for sensory input. While that description holds for certain *first-order* thalamic nuclei, *higher-order* nuclei receive substantial input from cortex itself and project broadly back to other cortical regions, forming *transthalamic pathways*. These cortico-thalamo-cortical circuits link primary to higher-order sensory cortices, motor cortices, sensorimotor regions, and frontal areas, offering a powerful alternative route of information flow across the brain. Evidence for transthalamic pathways comes primarily from mice, where their anatomy and function have been dissected in detail, but comparative data suggest that they are conserved across species and may represent a general feature of cortical organization. Initial work suggested that these pathways might be especially powerful because their

corticothalamic synapses are unusually strong compared to most corticocortical connections[16,17]. With the advent of viral tracing, calcium imaging, and optogenetic tools, research has now moved beyond synaptic physiology to direct tests of how transthalamic circuits contribute to behavior. These studies reveal surprising roles in decision-making, stimulus valuation, confidence, and predictive processing.

Transthalamic pathways arise from layer 5 (L5) pyramidal neurons of a "lower" cortical region and transmit through higher-order thalamus to "higher" cortical areas. L5b cells are major cortical output neurons that also target subcortical regions controlling movement[18–20], positioning transthalamic circuits as a hub for integrating motor, sensory, and cognitive signals. An important detail is that direct corticocortical and transthalamic circuits originate from largely separate neuronal populations in cortical L5[21], suggesting that they carry different streams of information. This begs the question: what information is sent through this indirect cortical circuit with a powerful synapse in thalamus and why?

In this review, we synthesize these circuit findings with a focus on their behavioral relevance but acknowledge the methodological limitations. We present emerging evidence that transthalamic pathways are well-positioned as dynamic integrators that transform and broadcast contextual signals, internal states, and task-relevant information across the cortex. By consolidating evidence across circuit physiology, systems neuroscience, and cognitive theory, we provide a conceptual update on thalamic function: from local communication to brain-wide computation.

[1]Department of Neurobiology, University of Chicago, Chicago, IL, USA. [2]The Florey Institute of Neuroscience and Mental Health, Melbourne, VIC, Australia. ✉e-mail: Christina.mo@florey.edu.au

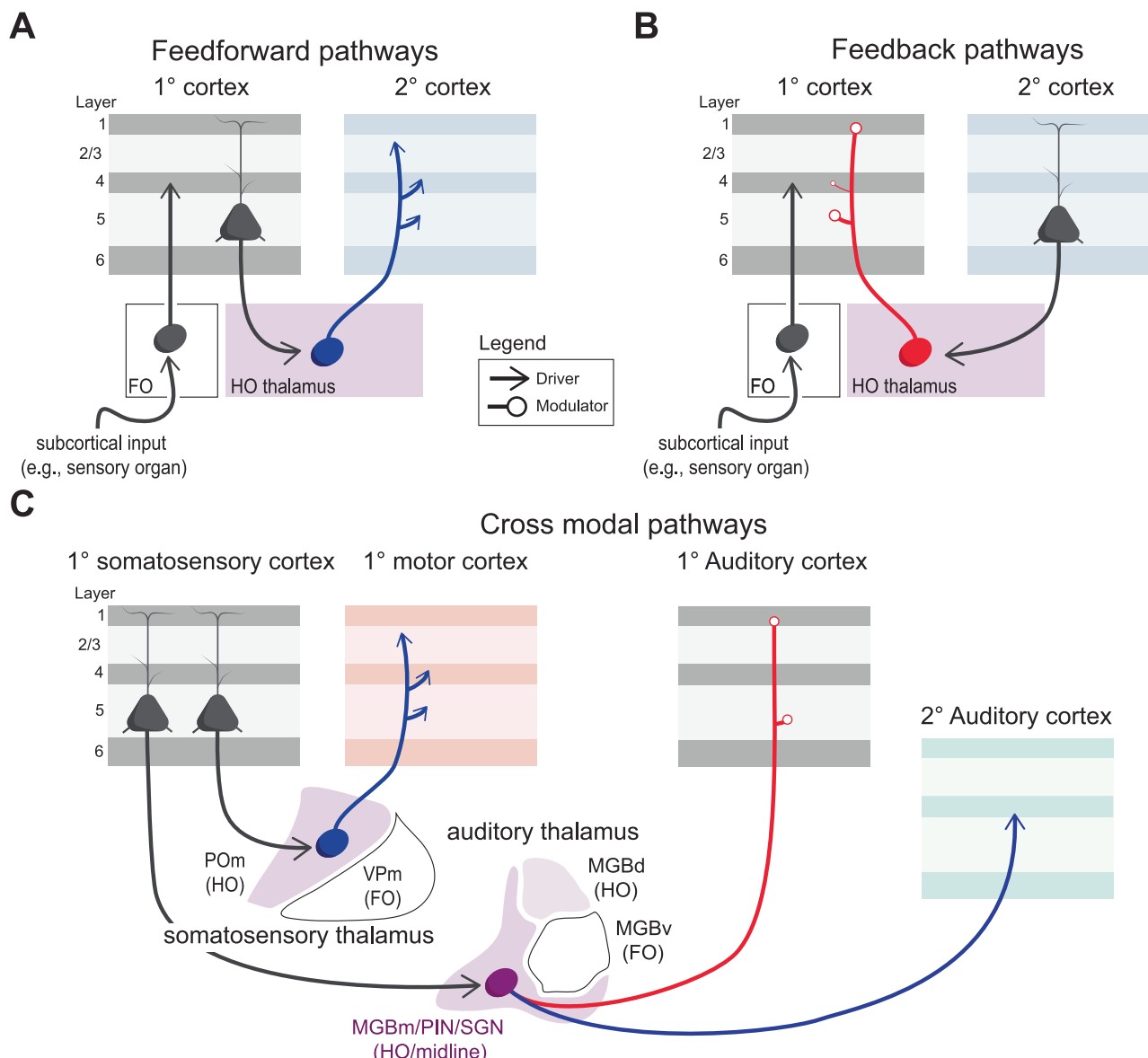

**Fig. 1 | Basic organization of transthalamic pathways. A** Feedforward transthalamic pathways arise from L5 of primary cortex to higher order cortical areas (e.g., secondary, tertiary, etc.) via HO thalamus. This pathway is comprised of all strong 'driver' synapses (arrows). **B** Feedback transthalamic pathways from L5 of higher order cortex project to HO thalamus via driver synapses but HO thalamic projections to primary cortex terminate with modulator synapses (open circles), particularly to L1 and L5. **C** Variants of this motif extend to motor and cross-modal circuits, such as S1–M1 (feedforward driver–driver) and S1–auditory cortex (driver–modulator).

## Transthalamic pathways: general wiring pattern

The prototypical feedforward transthalamic pathway was first demonstrated in the mouse somatosensory system, where activity in primary somatosensory cortex (S1) stimulated metabolic activity in the secondary somatosensory cortex (S2) via the higher-order thalamic nucleus, the posterior medial nucleus (POm)[22]. Subsequent studies have revealed similar feedforward pathways in the visual[23,24], sensorimotor[25] and motor systems[26].

Transthalamic circuits follow a conserved organization: layer 5 (L5) neurons in one cortical area send strong inputs to higher-order (HO) thalamus, which then relays similarly strong inputs to other cortical regions[16,24] (Fig. 1A). These strong connections at the synapse have been termed "drivers," and are specialized for reliably transmitting the excitatory sensory information, in contrast to the weaker "modulatory" connections that tune or adjust activity rather than carry the main signal[17,27]. Importantly, all known transthalamic pathways use driver-type connections, providing a

first indication of why they may exert such powerful influence over cortical processing.

By contrast, feedback transthalamic pathways (e.g., S2 → POm → S1) are organized differently: L5 inputs to HO thalamus are drivers, but HO projections back to primary cortex are modulatory, targeting mainly superficial and deep layers[24] (Fig. 1B). Considering the HO thalamic projections to a given cortical target are either driver to all layers or modulator to all layers, not mixed, it follows that standard transthalamic pathways conform a conserved organization shown in Fig. 1A, B[24,25].

Beyond sensory cortices, transthalamic circuits also link sensorimotor and motor areas[25] and even cross modalities (e.g., S1 → auditory thalamus → auditory cortex)[28] (Fig. 1C). This suggests a generalized circuit motif that connects primary to higher-order sensory cortices, motor and sensorimotor regions, and even frontal areas, and that is likely conserved across species, with the strongest evidence to date from mice.

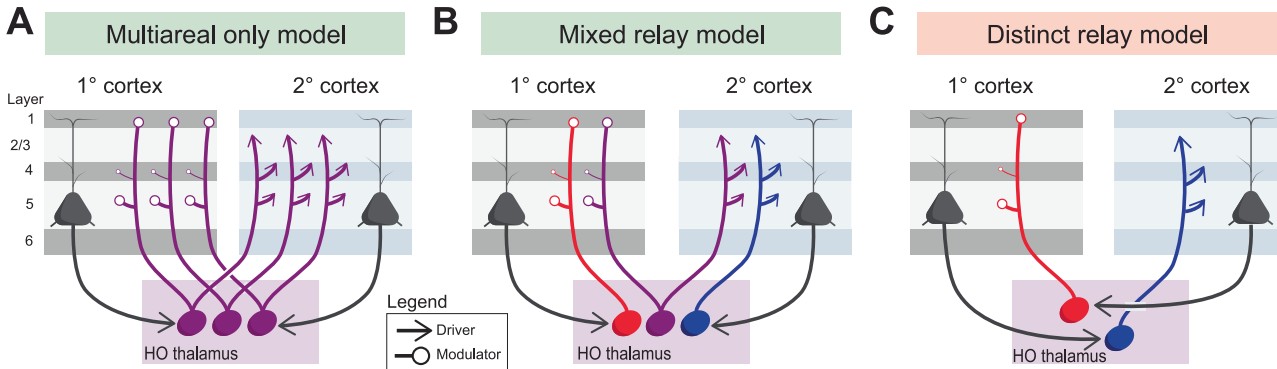

**Fig. 2 | How do individual thalamic relays contribute to transthalamic pathways?** **A** Most HO thalamic cells branch to innervate multiple cortical areas (multiareal, purple) and may therefore integrate both the feedforward and feedback pathways alone[34]. **B** An alternative model is that in addition to multiareal cells (purple), there are also separate feedforward and feedback transthalamic channels, mediated by two separate populations of HO thalamic cells (red and blue). **C** A final scenario is that the feedforward (blue) and feedback (red) pathways solely innervate distinct HO relay cells, effectively separating these two information streams. However, given the prevalence of multiareal cells[34] and degree of convergence of L5 inputs in HO thalamus[119], model C is unlikely (denoted by red shading over title), while models A and B are both plausible (denoted by green shading).

Notably, while we focus on transthalamic pathways, reciprocal connections between cortex and thalamus, sometimes called cortico-thalamo-cortical (CTC) loops, are also ubiquitous[29–31]. These are likely integrated with transthalamic pathways, as discussed in a recent review[32], but are not examined in detail here.

## How do individual thalamic relays participate in transthalamic pathways?

How HO thalamic neurons implement these circuits remains unresolved. Single-cell tracing in rodents using viral-based tracing and in vivo electroporation approaches shows that the majority of HO neurons branch to innervate, often densely, multiple cortical areas[33,34]. In other words, a large proportion of the neurons in HO thalamus are so called "multiareal cells"[34]. It follows that these cells would be capable of participating in both feedforward and feedback pathways simultaneously due to their projections to interlinked cortical areas - S1 and S2, for instance[33]. However, since the same HO nucleus (e.g., POm) provides driver inputs to some targets (e.g., S2) and modulators to others (e.g., S1), individual multiareal cells would need to manifest this functional divergence (i.e., drive one target but modulate another). Indeed, electron microscopy recently revealed single POm neurons innervating distinct cortices (M1 vs. S1) with markedly different axonal bouton morphologies[35] that correspond to their previously described functional roles as drivers or modulators[25]. Given the prevalence of multiareal neurons in HO thalamus, these cells may be the primary conduit of transthalamic information in both the feedforward and feedback direction (Fig. 2A).

Alternatively, subpopulations of HO neurons may specialize, such that some HO neurons project exclusively to primary cortex (modulators), others exclusively to higher cortex (drivers), and a third group—the multiareal cells branch to both (Fig. 2B). This mixed organization is most consistent with current data because, in addition to the substantial multiareal cell population in HO thalamus, there exist many cells that project to a single cortical target. This is evidenced both by bulk labeling, in which dual retrograde labeling of disparate visual cortices revealed only a fraction of HO thalamic neurons are doubly labeled (i.e. branch to multiple cortical areas; although this method is subject to false negatives)[36], and single-cell tracing, which demonstrates that a fraction of HO neurons (perhaps less than 20%) project to a single cortical area in restricted fashion[34,37]. These cells may also participate in transthalamic processing. A purely segregated model, where feedforward and feedback relays are entirely distinct, appears very unlikely given the prevalence of multiareal cells (Fig. 2C). Clarifying these contributions will be key for understanding how transthalamic signals are temporally and functionally distributed across the cortex.

## Difficulties in isolating the transthalamic pathway for study

Transthalamic pathways are disynaptic circuits (the corticothalamic and thalamocortical synapses) that run parallel with other corticothalamic (e.g., those from L6), thalamocortical, and corticocortical circuits. Therefore, selective study of the entire transthalamic pathway is very challenging and, to date, has only been achieved in studies using slice electrophysiology and anatomical tracing (including rabies-mediated transsynaptic tracing)[22–26]. These findings have revealed that feedforward transthalamic pathways are anatomically prominent and purely constituted by powerful driver synapses, which implicate a dominance in the awake state.

Experiments in awake animals thus far have only isolated one leg of the transthalamic pathway, leaving an open loop. For instance, silencing the cortical output from to HO thalamus indeed suppresses the first step, but while these projections to HO thalamus may synapse on cells that are transthalamic, they may also innervate those that project subcortically, such as the striatum or amygdala[38–40] (Fig. 3A). Thus, the observed behavioral effects cannot be entirely attributed to transthalamic function (Fig. 3A). Similarly, silencing the HO thalamic projections to cortex will impact the function of the second leg of the transthalamic pathway, but not all those HO thalamic cells receive input from cortical L5 (Fig. 3B).

One workaround is to measure how silencing this cortical L5→HO thalamus projection affects neuronal activity in a target region using, for instance, calcium imaging or multi-electrode recordings (Fig. 3C). Even so, indirect circuits between HO thalamus and cortex contribute to the observed effects. Combating the confound presented by targeting only the second leg of the transthalamic pathway (i.e., HO thalamus→cortex) requires the selective manipulation of HO relays that receive a L5 cortical input from the area of interest. However, plans to use anterograde, transsynaptic combinatorial viral techniques such as those used to isolate the subcortical projections to HO thalamus (i.e., AAV1-based approaches[41,42]) are thwarted by the fact that both L5 and L6 project to HO thalamus (i.e., transsynaptic transport will occur along both pathways, confounding the interpretation). Advances in anterograde tracing or combinatorial genetic approaches may provide fruitful solutions to this problem[43,44] (Fig. 3D).

In this review, we prioritize the discussion of experiments which record from, or perturb, specific transthalamic components, such as corticothalamic L5 projections to a defined HO nucleus (Fig. 3A), or thalamocortical projections from a HO nucleus to a specific cortical region (Fig. 3B). We exclude studies where entire cortical or thalamic brain regions were manipulated, as well as generalized L5 projections not targeted to HO thalamus. Although no study in behaving animals has yet fully isolated a complete transthalamic pathway, valuable insights have nonetheless emerged from work that interrogates its separate legs.

## Types of experiments discussed and their limitations

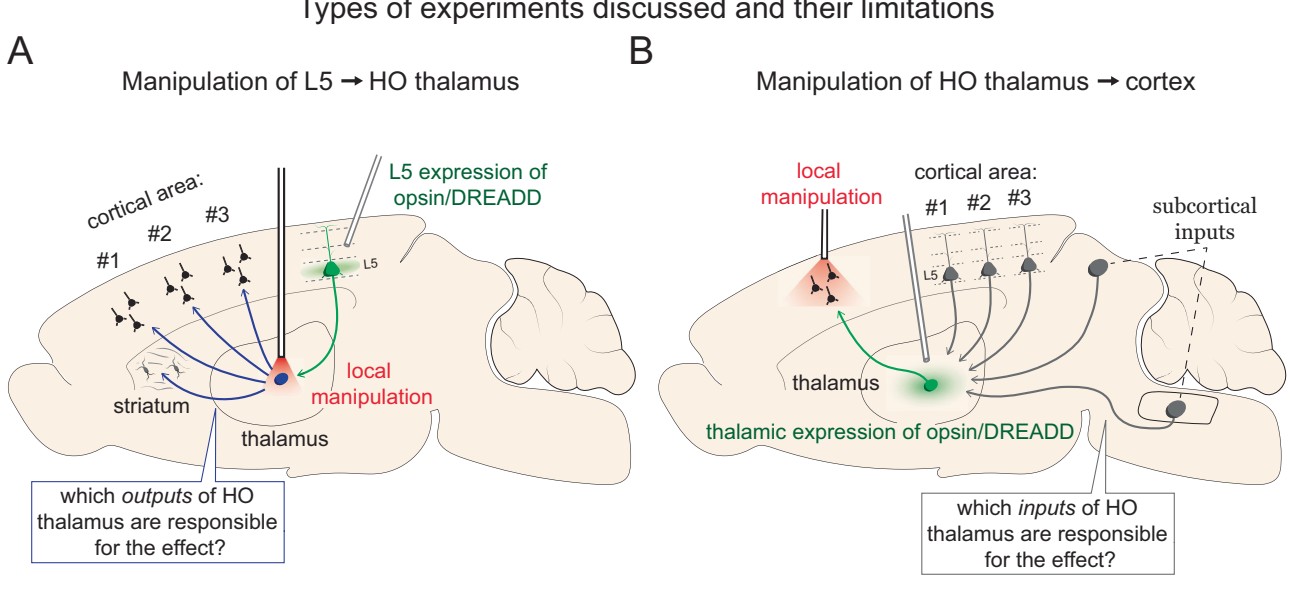

## Potential solutions to capture transthalamic pathways

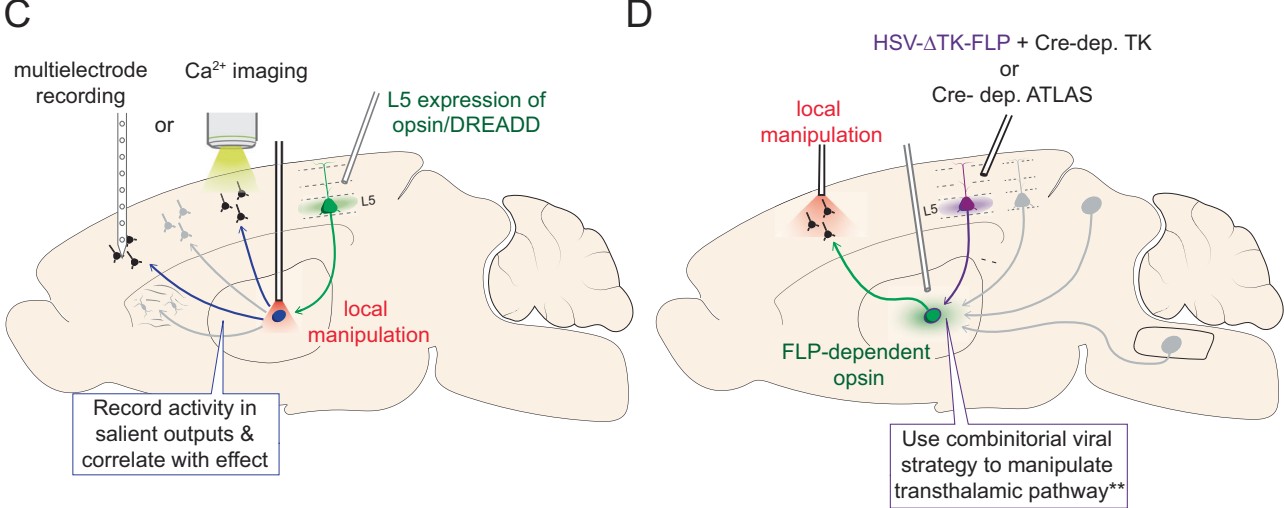

**Fig. 3 | Difficulties in experimentally manipulating the transthalamic pathway.**
**A, B** Depictions of the types of L5 corticothalamic and thalamocortical studies thus far and the potential confounds in interpreting their results as relying on transthalamic circuitry specifically. **A** Manipulating L5 inputs to HO thalamus affects downstream projections beyond cortex. **B** Manipulating HO thalamocortical projections may include non-transthalamic relays. **C** Combining manipulation with recording/imaging in downstream cortex provides a partial solution. **D** Next-generation viral and genetic strategies (**\*\*and a desirable toxicity profile), may allow isolation of the entire transthalamic pathway in future work[44].

## In vivo functional investigations of transthalamic pathway

Having established the organization of transthalamic pathways and methodological challenges, we examine recent studies in awake rodents that monitor or manipulate these circuits to understand their role in attention, decision making, and sensory perception (Table 1), which are operationalized by the observable outcomes in goal-directed stimulus-response tasks.

### Visual transthalamic pathways

The feedforward visual transthalamic pathway connects L5 of primary visual cortex (V1) to higher visual areas (HVAs) through the HO visual thalamic nucleus, the pulvinar (also referred to as lateral posterior nucleus in mice)[23,24,45]. Blot and colleagues first investigated these pathways in vivo, using two-photon calcium imaging to compare activity in corticothalamic versus thalamocortical projections to HVAs[23] (Table 1). Their results demonstrated that pulvinar-to-HVA boutons displayed response

preferences strongly correlated with L2/3 neurons in HVAs, whereas V1 corticocortical boutons had broader stimulus responses, indicating that pulvinar exerts a more targeted influence on HVAs than direct V1 input. The authors also manipulated the animals' visual flow by decoupling the virtual visual environment from treadmill activity. They found that pulvinar-to-HVA boutons encoded both optic flow and locomotor signals, while V1 corticocortical boutons primarily responded to optic flow. This suggests that HVAs rely on pulvinar input to integrate external visual signals with self-motion cues, and that this integration does not depend on direct V1 inputs[23].

Results from related studies by Han and Bonin (2024) and Neske and Cardin (2025) support that corticocortical and HO thalamocortical projections to HVAs carry distinct signals. Both studies also performed calcium imaging of L1 boutons from pulvinar to various HVAs in the awake mouse. Results from Han and Bonin largely corroborated with that of Blot et al., finding that the tuning of pulvinar→HVA boutons matches the tuning of the target HVA neurons, more so than the corticocortical inputs[46].

## Table 1 | Summary of studies examining transthalamic pathways in vivo

| Citation | Type of study | Pathway(s) (manipulated/ imaged) | Method of manipulation | Behavioral task/stimulus | Findings | Notes on specific study of the pathway |
|---|---|---|---|---|---|---|
| **Visual system** | | | | | | |
| Blot et al., 2021 | Visual stimulus plus state-related information and visual flow | Silenced: Visual cortical areas (V1, AL, PM) Imaged during silencing: Pulvinar→AL axons Pulvinar→PM axons | Excitatory opsin (ChrimsonR) expressed in interneurons (PV) of cortical region* | 1) Presentation of drifting gratings 2) Animal traverses a linear virtual track w/ or w/o decoupling from optic flow | 1a) V1 suppression strongly reduced pulvinar axon activity in AL and PM 1b) Pulvinar axons have tuning preferences that are similar to their target HVA 2) Pulvinar axons to HVAs carry both sensory and motor (self-motion) information, while intracortical connections are largely sensory | Compared corticocortical and thalamocortical inputs to higher order cortex *Cannot distinguish contribution of L5 and L6 with this approach. |
| Han and Bonin, 2024 | Visual stimulus only (no behavior) | Imaged: V1 and HVAs→AL V1 and HVAs→PM V1 and HVAs→A Pulvinar→V1 and HVAs | N/A | Presentation of a range of visual stimuli varying in spatiotemporal properties | a) Visual cortical areas with similar tuning preferences are more strongly anatomically interconnected b) Intracortical projections carry sensory information to their targets that is specific (i.e., matches the functional bias of the target), albeit relatively weakly. c) Pulvinar inputs to HVAs provide highly specific information to target HVAs (i.e., pulvinar input tuning preferences match their targets) | Compared corticocortical and thalamocortical inputs to higher order cortex |
| Neske and Cardin, 2025 | Visual stimulus plus state-related information | Silenced: Cortical or Pulvinar inputs to PM Imaged during silencing: Intracortical: V1→PM axons LM→PM axons Thalamo-cortical: Pulvinar→PM axons | Inhibitory opsin (eOPN3) expressed in thalamic or cortical inputs while imaging PM cell activity | 1) Presentation of drifting gratings. 2) Monitoring of movement and pupil size (arousal state). | 1a) Corticocortical projections carried sensory information to PM 1b) thalamocortical projections carried state-related information (e.g., related to pupil dilation) 2) Silencing intracortical (e.g., V1→PM) connections decreased contrast-response curves of PM neurons during passive presentation of visual gratings. Silencing thalamocortical projections only moderately decreased PM visual response curves | Compared corticocortical and thalamocortical inputs to higher order cortex |
| McKinnon et al., 2025 | Visual discrimination go, no-go behavioral task | Silenced: V1 L5 → pulvinar | Inhibitory opsin (Jaws) expressed in V1 L5 and terminals inhibited in pulvinar | Discrimination of oriented visual stimuli (static gratings) with psychometric testing | Silencing V1 L5 → pulvinar projections causes increased error rate, a flattening of the psychometric function, and reduced d-prime performance across all stimuli. | Manipulated only the corticothalamic leg of the pathway |
| **Somatosensory system** | | | | | | |
| Takahashi et al., 2020 | Whisker go, no-go detection task | Silenced: S1 L5→POm Imaged: Dendrites of S1 L5 cells | Inhibitory DREADD in S1 L5, local (POm) infusion of CNO | Detection of a single whisker movement, go/no-go task | Detection (or reporting) of stimulus is severely reduced with DREADDs inhibition* | * DREADD-based manipulation means that the targeted pathway was affected throughout all parts of the task. |
| Qi et al., 2022 | Whisker discrimination two-alternative forced choice task | Silenced: S1 L5→POm* | Inhibitory opsin injected into S1 L5 bilaterally and optic fibers implanted bilaterally over terminals in POm | Whisker discrimination task in freely behaving mice; detection of target texture accompanies water reward, 2-alternative forced choice task | Bilateral suppression of S1 L5 → POm pathway abolishes texture discrimination. Behavioral performance is reduced equally when whole POm is suppressed. | *We are highlighting only the transthalamic component; the authors also show that inhibiting M1/M2 L5→POm, and SpV→POm pathways has no behavioral effect. |
| Mo et al., 2024 | Whisker discrimination go, no-go task | Silenced: S1 L5→POm | Inhibitory opsin (Jaws) expressed in S1 L5 and | | Silencing S1 L5 →POm increased error rate, flattened the psychometric function | Study attempted to 'complete the circuit' by manipulating S1 L5→POm |

**Table 1 (continued) | Summary of studies examining transthalamic pathways in vivo**

| Citation | Type of study | Pathway(s) (manipulated/imaged) | Method of manipulation | Behavioral task/stimulus | Findings | Notes on specific study of the pathway |
|---|---|---|---|---|---|---|
| | | Imaged during silencing: S1 or S2 | terminals inhibited in POm while imaging S1 or S2 cell bodies | Texture discrimination and whisker detection tasks with psychometric testing; go/no-go task | of performance, and reduced d-prime performance across all stimuli, particularly during the sensory periods of the task. Reward-based neuronal discrimination is impaired by S1 L5 → POm inhibition, particularly in S2. | whilst imaging in S2 but POm→S2 was not directly studied. |
| **Motor system** | | | | | | |
| Takahashi et al., 2022 | Whisker deflection detection task | VM→ALM L1 | Inhibitory DREADD in VM, local (ALM) infusion of CNO; Excitatory ChR2 in VM, local LED stimulation in ALM | Manual deflection of whisker, licking to report | a) Licking/reporting of whisker detection detection is delayed with silencing of VM→ALM$_{L1}$ b) Licking/reporting of whisker deflection detection is accelerated with activation of VM→ALM L1 | Behavioral performance is preserved in these experiments—only the timing of motor response is affected. |
| **Prefrontal system** | | | | | | |
| Bolkan et al., 2017 | Spatial working memory task | MD→PFC MD→OFC | eArch3.0 in MD, local terminal suppression via optical fiber in PFC of OFC | Delayed nonmatch-to-sample T-maze task. | Suppression of MD→PFC projections, but not MD→OFC, decreases performance in T-maze task; suppression restricted to the delay phase (but not sample or choice phases) of the task accounts for performance decrease. | |
| | | mPFC→MD | eArch3.0 in PFC, local terminal suppression via optical fiber in MD* | | Suppression of PFC→MD pathway decreased task performance; suppression restricted to the either the delay or choice phases (but not sample phase) accounts for performance decrease. | *Cannot distinguish contribution of L5 and L6 with this approach |
| Alcaraz et al., 2018 | Instrumental learning task in rats | MD→PFC | Dual viral strategy using retrograde-Cre (Cav-2) in PFC and Cre-dependent hM4Di in MD | Instrumental learning task with specific reward tied to either level pressing or pushing of a tilt. Rewards were either devalued or degraded during test session | Suppression of MD→PFC pathway does not affect learning, but impairs behavioral flexibility of the animal during both devaluation of reward and degradation of the action-reward relationship | Separate study of corticothalamic and thalamocortical legs of the pathway |
| | | PFC→MD | Dual viral strategy using retrograde-Cre (Cav-2) in MD and Cre-dependent hM4Di in PFC | | Suppression PFC→MD pathway does not affect learning, and only affects choice behavior during devaluation of reward (not involved in action-outcome assessment) | |
| Lam et al., 2024 | Multisensory rule reversal task in tree shrews | Silencing: ACC→MD* ACC→PFC | iC ++ or halorhodopsin in ACC | Switching cued rules in a block design | Inhibiting ACC→MD delayed behavioural rule switching ACC → PFC corticocortical pathway had no effect | *Cannot distinguish contribution of L5 and L6 with this approach |

*A* anterior visual cortex, *ACC* anterior cingulate cortex, *AL* anterolateral visual cortex, *ALM* anterolateral motor cortex, *ChR2* channelrhodopsin, *CNO* clozapine-N-oxide, *DREADD* designer receptor exclusively activated by designer drugs, *FA* false alarm, *L* layer, *MD* mediodorsal thalamus, *PFC* prefrontal cortex; *PM* posteromedial visual cortex, *POm* posteromedial thalamus, *PV* parvalbumin, *S1* primary somatosensory cortex, *S2* secondary somatosensory cortex, *V1* primary visual cortex, *VGAT* vesicular GABA transporter, *VM* ventromedial motor thalamus.

Interestingly, this was not the case for pulvinar→V1 boutons, again emphasizing the feedback, modulatory function of HO thalamus to primary cortex[24]. In contrast, Neske and Cardin (2025) saw weak correlations between calcium activity in pulvinar→HVA boutons (area posteromedial, PM) and the PM cortical neurons they simultaneously imaged, but verified that pulvinar→PM inputs carry more arousal and movement information compared to corticocortical projections[47].

The aforementioned data were collected in the awake animal but in the absence of a decision task[23,46,47], leaving open questions about perceptual function. Addressing this, McKinnon et al. (2025) perturbed the corticothalamic leg of the transthalamic pathway during a visual discrimination task[48]. Mice were trained to lick in response to one visual stimulus (90° oriented grating) and withhold licking to another stimulus (0° oriented grating) for water reward. Specific optogenetic silencing of the V1 L5-to-pulvinar inputs impaired stimulus discrimination across a range of 0–90° orientations. In an important control, the behavioral effect was not seen when the stimulus was presented in visual space that activated the corresponding V1 area which did not express the inhibitory opsin[48]. This dependance on the corticothalamic projection suggests that the transthalamic pathway contributes to accurate perception during goal-directed behavior.

While these studies collectively provide excellent data for the likely roles of transthalamic pathways through pulvinar in visual perception, limitations remain. First, the two separate legs of the transthalamic pathway (corticothalamic and thalamocortical) were perturbed or examined across studies, but their results cannot be applied to understanding feedforward transthalamic pathways as a whole. As outlined in Fig. 3B, when assessing the functional implications of pulvinar versus corticocortical bouton activity in HVAs, the thalamocortical axons from pulvinar that are part of the transthalamic pathway (with inputs from cortical L5) cannot be distinguished from those that are driven by, for example, the superior colliculus, which provides distinct information about the visual scene and eye movements to pulvinar[38,49–52] (see also section "Integrating extrinsic signals with internal state").

Further, when assessing the functional contribution of transthalamic versus corticocortical pathways in higher visual areas, all studies imaged afferent boutons in L1[23,46,47]. This is an important detail, as all HO sensory nuclei studied to date project largely to the middle layers of higher-order cortical areas (some participating in feedforward transthalamic pathways)[24,25]. Meanwhile, projections to primary cortex (some comprising feedback transthalamic pathways) terminate with modulator synapses preferentially in L1 and L5[24]. Thus, the L1 bouton populations imaged in these studies may be enriched for particular subcircuits that fall outside the feedforward, all-driver organization that typifies transthalamic pathways. A more comprehensive capture of the laminar activity of the inputs to, and neurons in, V1 and HVAs is required to assess how sensory versus more complex, integrated information is distributed to visual cortex.

In sum, these in vivo studies demonstrate the existence of a feedforward visual transthalamic pathway through pulvinar[23] that is required for visual discrimination[48]. They also show that signals from this HO nucleus, some of which are likely to be driven by the transthalamic circuit, contribute to disambiguation between self-motion and visual signals of the corticocortical projection[23,46,47] (see further discussion in Section "Integrating extrinsic signals with internal state").

### Somatosensory transthalamic pathways

The somatosensory feedforward transthalamic pathway links S1 L5 to secondary somatosensory cortex (S2) via the higher-order thalamic posterior medial nucleus (POm)[22,53]. Inhibiting this first leg of the pathway, S1 L5→POm, leads to robust deficits in whisker-based somatosensory tasks (Fig. 3A). Takahashi and colleagues used chemogenetics to specifically silence S1 L5 terminals in POm (Table 1), which severely diminished the ability to detect whisker deflections[12]. Targeted silencing of other S1 L5 outputs to the superior colliculus and striatum also disrupted performance

but effects were weaker or absent for brainstem targets. In complementary experiments, increases in S1 L5 apical dendritic calcium activity predicted detection of the whisker deflection. This suggests that S1 L5 outputs to multiple subcortical structures, including POm, provide essential information for perceiving and reporting the presence of a somatosensory stimulus.

Using a complementary technique, Qi et al. (2022) optogenetically inhibited various projections to POm during a freely moving texture discrimination task in mice[54]. Silencing brainstem projections from the spinal nucleus of the fifth nerve (SpV), which transmits whisker (and facial) sensory information to POm, had little effect on texture discrimination. Similarly, suppression of L5 inputs to POm from primary and secondary motor cortex (M1 and M2) did not affect performance. In contrast, silencing S1 L5 terminals in POm reduced discrimination levels, most effectively on trials that were more difficult to discriminate. These data highlight the unique importance of S1 L5 to POm to discrimination ability, a function that relies on intact S1[55].

Further refining these findings, Mo et al. selectively inhibited the S1 L5 to POm projection during specific task epochs, revealing that pathway activity during sensory sampling was essential for discrimination, whereas inhibition during the delay period increased discrimination thresholds without abolishing performance[53]. Notably, discrimination ability was impaired across texture difficulties to a level similar to performance in the absence of whiskers, and the degree of impairment correlated with the estimated magnitude of opsin activation[53]. These results indicate that transthalamic pathway activity is essential for aspects of behavioral performance that extend beyond the initial encoding of sensory stimuli (but that do not include the motor action itself). This is discussed further in Sections "Stimulus detection and discrimination" and "Maintenance of a percept".

A limitation of the neural silencing paradigms shared by all these studies[12,53,54] is that there are multiple downstream impacts of inhibiting S1 L5→POm which could underlie the behavioral deficits (Fig. 3A). POm projects not only to higher-order somatosensory and association cortices[56], but also motor cortex[25], striatum[40,57], and in feedback fashion to S1, where it can modulate future driver responses through mGluR activation[24]. Thus, the behavioral deficits reported likely reflect the combined impact of reduced activation in multiple circuits.

To probe this, Mo and colleagues (2024) combined inhibition of S1 L5→POm terminals with two-photon calcium imaging in S1 and S2, thereby recording impacts on the feedforward transthalamic pathway to S2 (as in Fig. 3C) and also the reciprocal pathway back to S1. Responses in S1 and S2 showed higher response selectivity for the rewarded ('hit') texture over the unrewarded ('correct-rejection' (CR)) texture (Fig. 4A). This texture selectivity is a key correlate of expert discrimination performance and successful reversal learning[58–60]. Silencing S1 L5 → POm inputs during sensory sampling did not change overall texture responsiveness in S1 or S2. Rather, it reversed selectivity of the S2 population, in alignment with increased performance errors (Fig. 4B). In S1, inhibition changed cell discriminability to equal fractions of those selective for the hit and CR textures (Fig. 4B)[53]. This more modest impact of inhibiting S1 L5 → POm on S1 responses is expected based on the modulatory synaptic properties of the POm → S1 projection[61]. In contrast, the substantial loss of selectivity in S2 cells is expected based on the driver properties of the POm → S2 projection[24,61]. However, the involvement of the direct POm→S2 projection was only implied, and further exploration is needed into yet-to-be-established intermediaries (e.g. striatum) to completely explain the results.

In summary, across several studies, silencing the first leg of the transthalamic pathway (S1 L5→POm) severely impaired whisker-based detection and discrimination performance[12,53,54] with the sensory period of the task particularly germane to behavioral success. Together, these findings highlight the transthalamic route as a critical driver of perceptual performance, extending beyond local encoding in S1 to shape distributed cortical representations in S2[53].

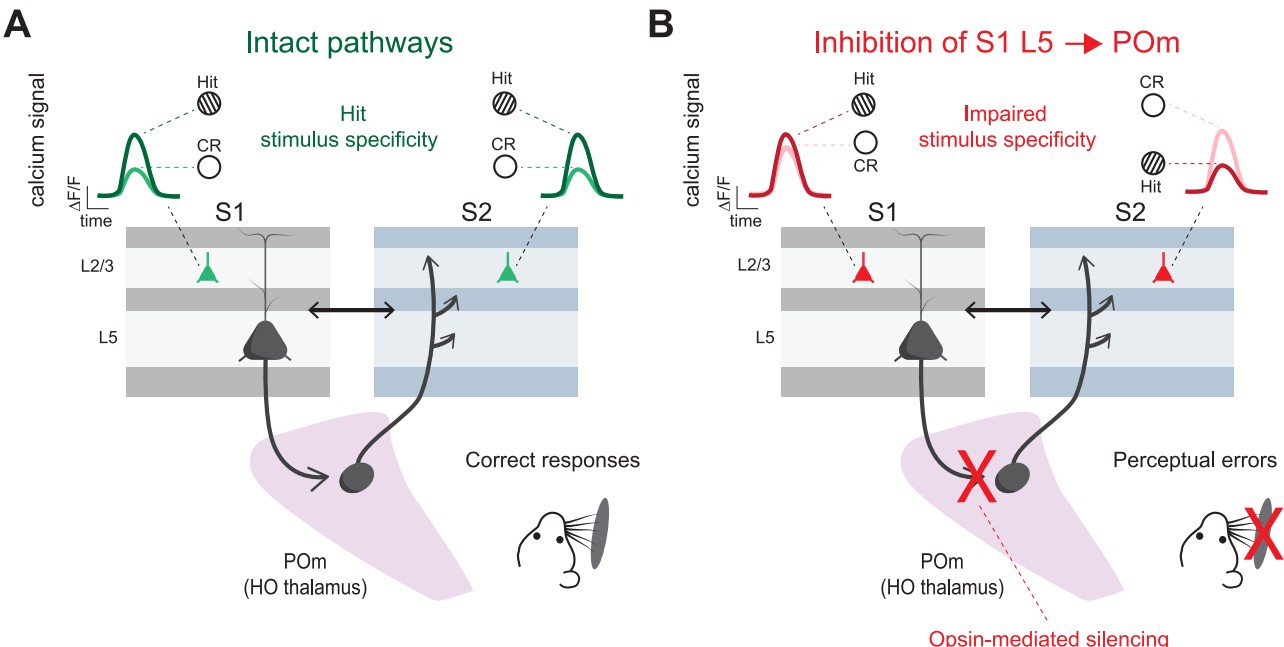

**Fig. 4 | Feedforward transthalamic pathway through POm encodes stimulus preference. A** In expert mice trained to discriminate between a rewarded texture (hit texture) and an unrewarded texture (CR texture), there is hit stimulus selectivity in S1 and S2 neurons, recorded in L2/3. **B** When the first leg of the somatosensory transthalamic pathway (S1 L5 → POm) is inhibited at the synapse in thalamus, the textures are no longer distinguishable in S1 (likely via S1 L5 → POm → S1) and reversed in S2, aligning with discrimination errors. Adapted from[53].

## Auditory transthalamic pathways

Anatomical studies have identified an auditory transthalamic pathway linking primary to secondary auditory cortex via the higher-order thalamic nucleus of the auditory system, the dorsal division of the medial geniculate body (MGBd)[62]. In particular, MGBd is suggested to act as a transthalamic node for corticocolliculo-thalamocortical signaling[28,63]. However, functional studies remain scarce. Lohse et al. (2021) identified a cross-modal pathway from S1 L5 to medial auditory thalamic nuclei, modulating auditory cortical activity[28]. The role of MGBd in auditory transthalamic processing and its behavioral significance remain unexplored.

## Motor transthalamic pathways

While sensory transthalamic pathways are increasingly well-characterized, much less is known about their counterparts in motor systems. Understanding these pathways is important because they may provide a route for integrating cortical and subcortical signals that guide the initiation and vigor of movement. The motor thalamus is a conglomerate of the ventromedial (VM), ventroanterior (VA), and ventrolateral (VL) nuclei, which are reciprocally and non-reciprocally connected to motor cortical regions[64,65]. A recent study demonstrated that M1 L5 innervates VA/VL cells with driver type synaptic properties and established a feedforward motor transthalamic pathway from M1 L5→VA/VL→M2[26]. Furthermore, there appears to be a small amplitude input from M1 onto a fraction of VM→ALM relays, demonstrating a minor feedforward transthalamic pathway through VM[31].

The thalamocortical leg of a motor transthalamic pathway has been functionally characterized in a study by Takahashi et al. (2021). During a whisker deflection task, the authors imaged and manipulated the L1 bouton activity of VM-to-ALM projections (as in Fig. 3B). They demonstrated that the thalamocortical axons from VM→ALM L1 were activated coincident with the initiation of lick responses, and performed chemogenetic inhibition or optogenetic activation to show that the projection is important for learned movement initiation[66]. However, the VM integrates inputs from several cortical and subcortical areas, including the basal ganglia, which is known to contribute to movement urgency and vigor[67] (as in Fig. 3B). Therefore, as with most studies discussed herein, future experiments will

need to selectively assess the contribution of the transthalamic circuits to generating or maintaining movements.

## Frontal transthalamic pathways

Transthalamic pathways through frontal thalamic nuclei are of particular interest because they link prefrontal cortex with other cortical regions and may support higher cognitive functions such as working memory, flexibility, and adaptive decision-making. These circuits offer a potential mechanism by which thalamus contributes to cognition beyond simple sensory relay. The mediodorsal nucleus (MD) of the thalamus receives a driver-like input from L5 of several frontal, sensory, and motor cortical regions[29,68–70], and projects to frontal cortical areas[71]. As such, MD is considered the HO thalamic nucleus subserving functions associated with prefrontal cortical areas[72].

So far, however, direct evidence for MD forming a feedforward cortico-thalamo-cortical pathway between two distinct cortical areas remains limited. One candidate pathway has been implicated from ACC → MD → PFC[73], but most studied connections appear largely reciprocal. Moreover, MD→frontal cortex terminations may not be classical drivers, with relatively small presynaptic boutons in L1, L3 and deep layers of prefrontal cortex[74,75]. Although, frontal cortical regions in rodents largely lack a thalamocortical recipient L4[76], suggesting the dense plexus of MD terminals in L3 might represent the equivalent feedforward projection from HO thalamus to higher order cortex in sensory systems. Considering transthalamic pathways through MD are currently understood as reciprocal connections, we discuss these studies here. Future investigations will reveal potential feedforward pathways from sensorimotor cortical areas with which MD is connected.

Functional manipulations of MD-PFC circuits highlight their role in cognitive flexibility, working memory, and rule learning[73,77,78]. For instance, one study silenced the activity of MD and mPFC circuits trained to traverse two T-maze arms sequentially, separated by a delay[77]. The MD→PFC pathway was required during the delay phase, and mPFC recordings showed that delay-period activity depended on MD input. These findings corroborate similar data using a split-attention task[79], demonstrating that

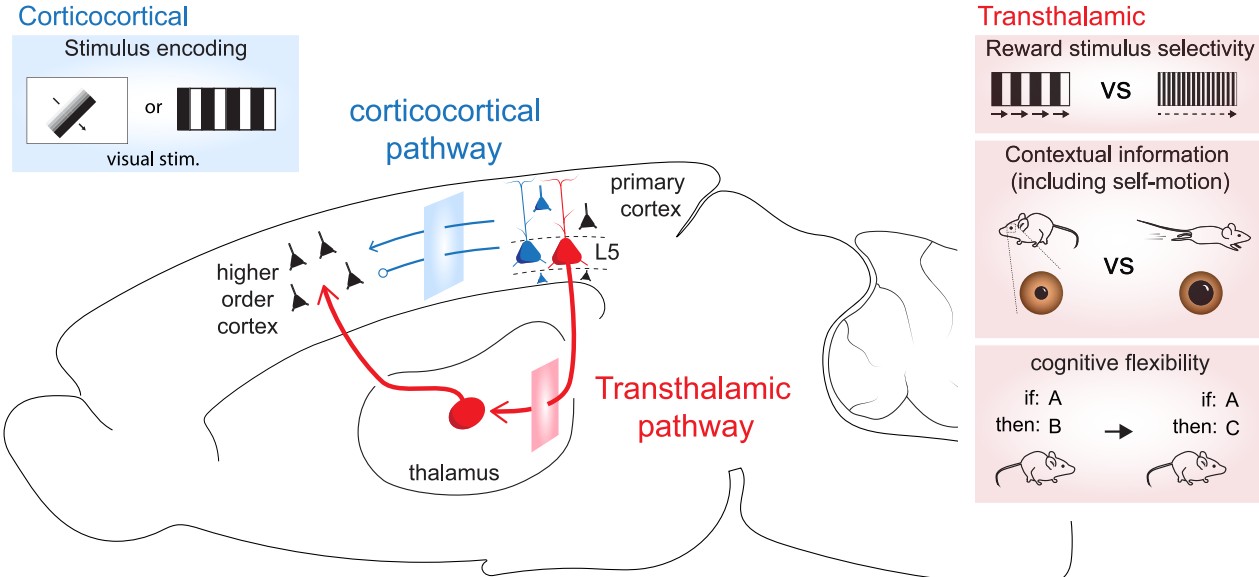

**Fig. 5 | Summary of transthalamic pathway functions to date.** Compared to corticocortical pathways, which primarily transmit information about sensory stimuli, transthalamic pathways integrate sensory information with learned and internally generated signals. These include reward-related stimulus response preferences, sensorimotor signals associated with self-generated movements (i.e., efference copies), and internal state information such as arousal and movement. The frontal transthalamic pathways studied thus far enable the updating of action-reward and -error outcomes.

the MD→PFC pathway is crucial for trial-by-trial learning and short-term rule maintenance[77,80].

Other studies dissected the bidirectional MD–mPFC pathway during action-outcome learning (Table 1). Inhibition experiments revealed that corticothalamic projections contributed to reward-value assignment, while the MD → mPFC pathway was required for updating action-outcome contingencies[81]. Two studies build on these data using a complex multi-sensory task design with visual or auditory rulesets[73,78]. Suppression of the MD→PFC pathway impaired the ability to rule switch in response to the initial cue. Suppressing MD required twice the number of trials to switch and update their choices[78]. Lam et al. (2024) demonstrated in tree shrews that optogenetic inhibition of ACC-to-MD projections disrupted a similar adaptive decision-making task, implicating a transthalamic pathway between frontal cortices[73]. However, whether these connections form a true feedforward transthalamic circuit remains to be determined.

Together, these studies demonstrate that MD→PFC circuitry is required to support behavioral flexibility in adaptive decision-making tasks[77,78,81] with potential transthalamic inputs from PFC itself or the ACC[73]. This idea converges with findings in monkeys and humans that MD function is required for aspects of cognition and specifically implicate the MD→PFC pathway in maintaining task-rule associations across a delay[68,71]. Yet, the circuit-level details of cortico-thalamo-cortical interactions through MD require further scrutiny to isolate the particular roles of transthalamic pathways in these behaviors.

## Proposed functions of transthalamic pathways

Recent behavioral studies have shed light on the functional roles of transthalamic pathways, albeit by analyzing the two parts separately (Table 1). Here, we consolidate these insights into broader functional motifs and consider how they align with earlier hypotheses about cortico-thalamo-cortical signaling (e.g.[82]).

### Stimulus detection and discrimination

A well-substantiated role for transthalamic pathways is sensory processing in learned responses (Fig. 5). This is most clearly shown in the somato-sensory and visual systems, where inactivation of the S1 L5→POm or V1 L5→pulvinar pathways impairs discrimination of textures and gratings,

respectively[12,48,53,54]. Notably, inhibiting S1 L5→POm strongly disrupted texture discrimination but had only mild effects on simple detection[53,54], consistent with prior work showing that S1 is dispensable for detection but critical for fine discrimination[55,83]. In the visual system, pulvinar-to-HVA projections encode stimulus features that match the preferences of target cortical neurons, implying that HO thalamus contributes to shaping stimulus selectivity in higher visual cortex[23,84]. Additionally, McKinnon et al. (2025) showed that V1 L5 input to pulvinar is necessary for discriminating visual gratings, further supporting a role for transthalamic pathways in feature preference encoding.

Transthalamic pathways also appear to carry reward-related information that shapes sensory selectivity. Neurons in S1 and S2 preferentially respond to rewarded stimuli, and this preference emerges over learning[58–60]. Reward signals in S1 L5 neuron dendrites contribute to perceptual learning[85–87], while suppression of S1 L5→POm diminishes reward preference in S2 neurons[53]. Similarly, POm and pulvinar neurons robustly respond to the behavioral salience of stimuli, regardless of the sensory stimulus modality[88,89]. Taken together, these studies suggest that transthalamic pathways integrate both sensory features and reward relevance, helping shape perception in a way that direct corticocortical projections may not. Exactly what information is encoded and computed at the level of the higher-order thalamus, remain key questions for future work.

### Maintenance of a percept

Persistent neural activity has long been proposed as a mechanism for perceptual stability[90], especially in thalamo-cortical circuits[91,92]. While pre-frontal thalamo-cortical circuits are crucial for working memory[78,93], evidence for sensory thalamocortical contributions remains limited[94]. A modest deficit in discrimination performance was observed when the S1 L5→POm pathway was silenced during a delay period[53], hinting that sensory transthalamic circuits may help maintain stimulus information across short timescales. However, the inhibition could have suppressed the transmission of transthalamic signals to target areas, such as motor cortex[5,6,25] or even striatum[40,57], which in turn interrupted stimulus-response or motor processing. More selective perturbations will be required to test whether sensory transthalamic pathways truly support stable percepts, as has been proposed for frontal thalamocortical circuits.

### Integrating extrinsic signals with internal state

A key distinction of transthalamic pathways compared to corticocortical routes is their integration of external sensory inputs with internal state information. For example, pulvinar inputs to higher visual cortex correlate with arousal, indexed by pupil diameter[47], and POm/pulvinar activity scales with arousal levels more broadly[95]. The source of these state-dependent signals remains unclear but may involve neuromodulatory inputs (e.g. acetylcholine), since they act directly on HO thalamic cells[96], and on GABAergic inputs to HO thalamus such as the zona incerta[97], or cortical projection neurons that themselves encode arousal[47,98]. By transmitting these combined signals to higher cortical areas, transthalamic circuits may enable sensory processing to be dynamically tuned by behavioral context, a key area for future investigation.

### Distinguishing external and self-generated motion

Another proposed function of transthalamic pathways is the transmission of efference copies: predictive motor signals that help the brain distinguish self-generated from external stimuli[99]. This idea is supported by the anatomy of L5 corticothalamic neurons, which collateralize to motor centers in the brainstem or spinal cord[18,20,37,100–102]. This organization contrasts that of the separate population of corticocortical-projecting L5 neurons[21,103]. Activation of L5 projections in primary cortex can directly induce movements via brainstem sites[104]. Since the same action potential pattern extends to all axonal branches, it has been posited that transthalamic circuits convey predictive motor signals to sensory cortex, aiding in distinguishing self-generated from external stimuli[82]. Considering that this hypothesis has been discussed in detail previously[16], we focus here on the recent in vivo studies of transthalamic circuitry, which generally support it (however, see ref. [95]).

In line with this, recent studies report visuomotor mismatch signals carried by pulvinar-to-cortex projections when expected sensory consequences of movement are violated[23,105]. Pulvinar inputs to V1 also encode saccade-induced motion[106] and amplify unexpected visual flow through a disinhibitory cortical circuit[107]. These findings suggest a key role for transthalamic pathways in distinguishing environmental stimuli from self-generated changes[23,105,106] (Fig. 5), though direct tests of efference copy transmission via relevant tasks with detailed movement tracking, await future testing.

### Comparison of corticocortical and transthalamic pathway function

As introduced, the impacts of cortical inactivation during cognitive tasks have often been interpreted as a result of disrupted direct corticocortical processing. However, recent work has shown that cortico-subcortical signaling, rather than corticocortical signaling, supports sensory decision-making[12,13,108]. Takahashi and colleagues (2020) compared the roles of corticocortical S1 (intratelencephalic)-projecting neurons with S1 cortico-subcortical (pyramidal tract L5)-projecting neurons during a whisker detection task. While optogenetic activation of cortico-subcortical S1 dendrites increased the detection of a whisker movement, the same manipulation of corticocortical dendrites had no effect[12]. In a related study, the activity of these pathways was investigated across associative learning, finding that cortico-subcortical-projecting dendrites showed increasing reward-associated activity with learning, while corticocortical-projecting dendrites in S1 signaled stable, sensory representations[87]. Indeed, in a texture discrimination study, the dendritic activity of S1 L5 → S2 neurons did not reflect the improvement in discrimination ability over learning, but their inhibition during the outcome period blocked learning, suggesting a role in integrating outcome salience[86]. Work in the auditory system also showed that L5 subcortical-projecting cells more strongly represent stimulus-to-choice coupling compared to corticocortical-projecting cells[108]. Accordingly, in a separate study using an auditory decision task based on evidence accumulation, inhibition of subcortical-projecting neurons in the parietal cortex (during either the stimulus presentation or delay periods) impaired decision-making, whilst silencing corticocortical-projecting neurons did not[13].

Taken together, there is mounting evidence that cortico-subcortical projections from L5 are major determinants of perception, particularly the integration of the rewarded stimulus with action[12,13,87,108] whilst corticocortical projections to higher-order cortex represent sensory feature information[109–111], although stimulus-reward association activity develops with learning[87,112,113]. These investigations of cortico-subcortical projections in general likely apply to the first leg of transthalamic signaling, which emanates from cortical L5 to HO thalamus, and the suppression of which leads to large deficits in perceptual performance[12,53,54] (see Section "Somatosensory transthalamic pathways").

Regarding the second leg of transthalamic pathways, HO thalamus appears to exert a stronger influence on cortical function than direct cortical inputs. For instance, thalamocortical projections to frontal cortex contribute more significantly to stimulus processing, choice, and response selection than corticocortical inputs[114]. In the visual system, HO thalamocortical projections shape neuronal selectivity more than cortical inputs to a target higher visual area[23,84]. Collectively, these findings suggest that transthalamic pathways serve as strong conduits for sensory-motor communication, requiring that they be considered at least as deeply as direct corticocortical interactions in future models of global cortical function[11,115] (Fig. 5). However, as discussed previously, we await technology that allows manipulation of the complete pathway spanning the three brain regions to make more concrete predictions of transthalamic function. It is important to note that selective silencing of the direct corticocortical pathway is also rare[116], which emphasizes the need for more scrupulous dissection of circuitry if we are to accurately assign function.

## Future perspectives
### Modulation and gating of transthalamic pathways

One emerging view is that transthalamic pathways dynamically modulate or gate information flow between cortical regions in ways that corticocortical circuits cannot (Fig. 6A). While HO thalamic nuclei receive strong driver inputs from L5, they also integrate diverse subcortical inputs[51,117,118]. For instance, POm integrate afferents from L5 of somatosensory and motor cortex, the spinal nucleus of the trigeminal nerve (SpV), dorsal column nuclei, superior colliculus, and inhibitory inputs from zona incerta and anterior pretectal nucleus[117,119,120]. Physiological data now demonstrate that this integration occurs on the level of individual POm neurons[121]. This convergence enables individual thalamic neurons to act as computational gates, flexibly transmitting top–down, bottom–up, or multimodal signals depending on their timing[122] (Fig. 6B).

In contrast to converging excitatory streams, GABAergic inputs from the extrareticular GABAergic nuclei or basal ganglia may selectively suppress information traversing HO thalamus (Fig. 6B). In this way, GABAergic inputs to a given transthalamic relay can dynamically determine which cortical areas are connected by transthalamic circuits and which are not, as discussed recently[123] (Fig. 6B).

In addition to active shunting, the properties of the GABAergic neurons impinging on HO relays suggest that HO thalamus is at least partially governed by a disinhibitory functional motif (Fig. 6C–F). The basal ganglia, zona incerta, and a subpopulation of neurons in the anterior pretectal nuclei fire tonically at reasonably high rates at baseline, thereby providing a tonic inhibition to HO relays[124–127]. Upon release from this robust suppression, HO relays are temporarily disinhibited and more likely to fire in response to an incoming excitatory input. Such a mechanism has been demonstrated—specifically, disinhibition of POm relays is selectively gated by motor cortical L5 inputs to zona incerta, which act locally to suppress neighboring incertal cells and thereby release POm relays from tonic inhibition[128,129] (Fig. 6C, D). A similar organization appears in the motor thalamus via basal ganglia interactions[26] (Fig. 6E, F). Future behavioral experiments that silence the subcortical circuits to HO transthalamic relays will provide crucial insight into how these pathways are regulated by endogenous mechanisms, and for what perceptual function.

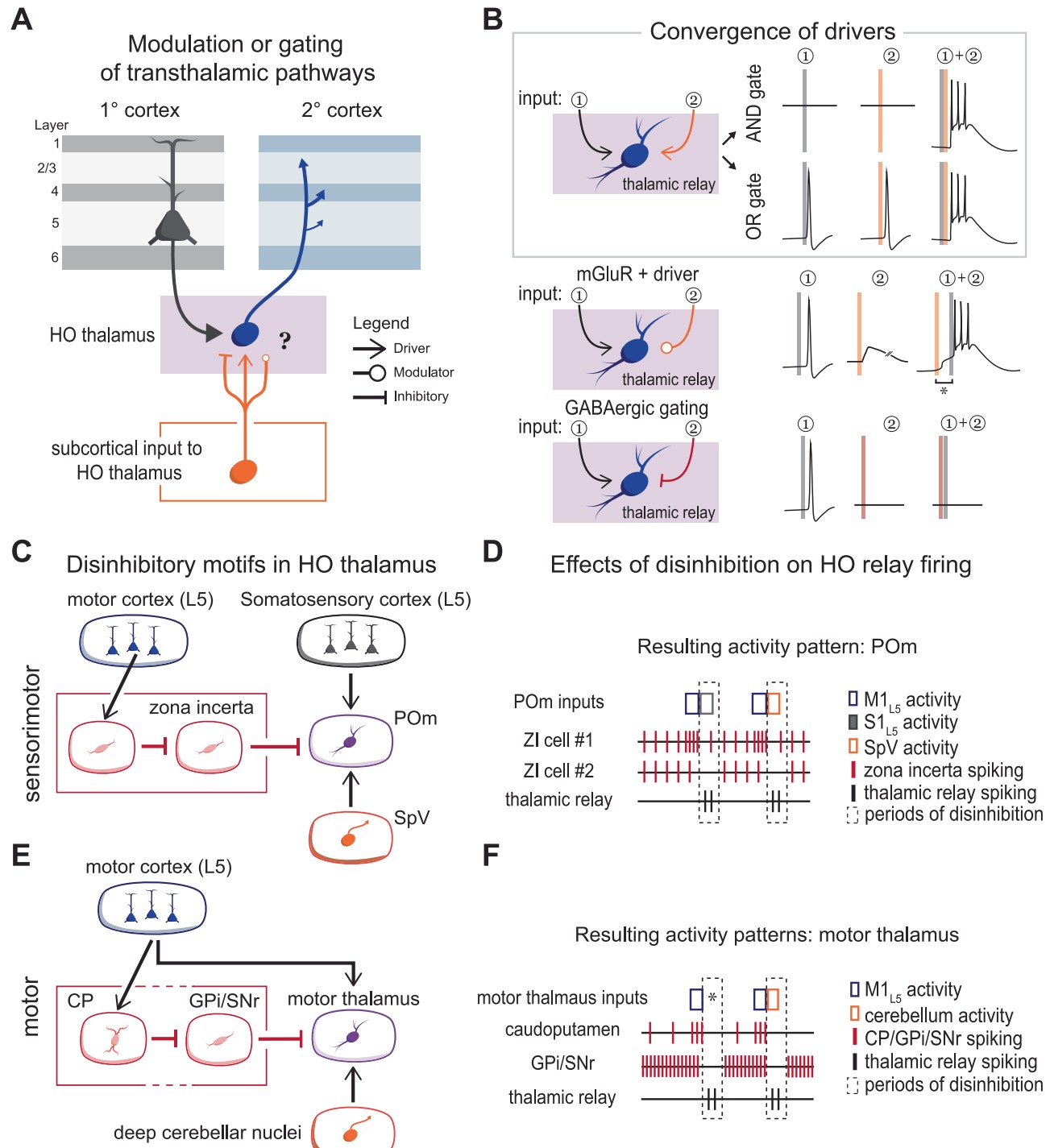

**Fig. 6 | Modulation, gating, and disinhibition of transthalamic pathways. A)** Secondary inputs to HO thalamus can modulate or gate the function of transthalamic circuitry, depending on the type of input (i.e., drivers, modulators, or inhibitory/GABAergic). **B** Convergent driving excitatory signals, such as from SpV and S1 L5, may allow HO thalamic relays to act as AND or OR gates[122]. Similarly, mGluR activation from ascending modulatory inputs might allow a potentiation of subsequent driving responses over the extended time frame of mGluR-dependent depolarization (denoted by asterisk). On the other hand, inhibitory inputs (e.g., from zona incerta) may gate excitatory information flow, as recently demonstrated[26] and discussed[123]. Importantly, the propensity of GABAergic inputs to HO thalamus from cells with a high baseline firing rate (20-50+Hz) sets the stage for a disinhibitory motif across HO thalamic nuclei. **C** L5 of motor cortex can govern POm relay cell firing by driving a local disinhibitory circuit via the zona incerta (red outline)[129].

Presumably, excitatory drive to disinhibited cells can arise from bottom–up (SpV) or top–down (somatosensory cortex) sources. **D** Specifically, activity from the motor cortex drives the activity of zona incerta cells (ZI cell #1) that inhibit nearby POm-projecting incertal cells (ZI cell #2), creating a disinhibitory window for excitatory activation by any input to POm. **E** In the motor thalamus, L5 projections from motor cortices branch to innervate both the basal ganglia and motor thalamus. Therefore, cortical signals might disinhibit motor thalamus through a circuit traversing the caudoputamen and internal globus pallidus (GPi) or substantia nigra pars reticulata (SNr). **F** Motor thalamic disinhibition is achieved via the basal ganglia. Disinhibited motor thalamic relays can be driven by bottom–up inputs from the deep cerebellar nuclei or L5 of motor cortex. Under some circumstances (denoted by an asterisk in **F**) disinhibition alone might drive relay cell firing due to t-type calcium channel-mediated rebound spiking.

### Transthalamic pathways in sensation, versus motion or cognition

Most work on transthalamic circuits so far has examined sensory and cognitive processes (Table 1)—whether the same organizational principles extend to motor and frontal systems is unclear. For instance, while sensory information reaches primary sensory cortex and ascends the cortical hierarchy[130,131], motor commands emerge in what are considered premotor or higher order cortical areas prior to movement initiation in primary motor cortex[31,132], potentially reversing this hierarchy.

A second potential point of distinction lies in the cortico-thalamo-cortical connectivity pattern in sensory versus motor areas. While transthalamic circuits in frontal and motor areas appear to be dominated by reciprocal connectivity loops, sensory transthalamic circuits demonstrate a propensity for forming feedforward and feedback pathways in addition to reciprocal ones[29,31,45]. In addition, with the diversity of thalamocortical targets, the functional consequences of transthalamic circuit activation can be expected to vary based on the cortical circuitry. For instance, while HO somatosensory thalamus output HO somatosensory cortex appears to largely avoid interneurons, the innervation pattern in primary sensory cortex[133], and perhaps in motor cortex[134], encompasses several interneuron subtypes. Therefore, studies examining the population activity in these cortical regions should take into account the synaptic targets and local circuitry. These variations suggest that transthalamic pathways are a shared motif with system-specific specializations, tailored to the computations of each domain.

### Development and plasticity of transthalamic pathways

This review has focused solely on the function of transthalamic pathways in adult animals, where such circuits are presumably fully mature. In contrast, how transthalamic pathways develop remains largely unexplored. Although the development of thalamocortical and layer 5 (L5) corticothalamic projections has been studied extensively[135–138] it is unclear whether coordinating a transthalamic pathway across two cortical targets (sometimes spanning distal regions such as sensory and frontal cortex), requires additional developmental constraints beyond those governing reciprocal cortico-thalamo–cortical loops. Another wrinkle is related to how HO neurons participate in transthalamic circuits (Section "How do individual thalamic relays participate in transthalamic pathways?"): If it is true that transthalamic circuits are largely facilitated by certain cell types, such as multiareal cells, their distinct developmental profile (e.g., molecular identity)[34] might allow for appropriate targeting by those L5 inputs that will ultimately form a transthalamic circuit.

Transthalamic circuits also exhibit clear short-term synaptic dynamics. L5 inputs to HO neurons display strong paired-pulse depression characteristic of driver synapses and HO intrinsic properties enable switching between burst ("wake-up call") and tonic firing modes ("faithful information transmission")[139–142]. These mechanisms are well described[143,144], but their behavioral relevance within intact transthalamic circuits remains incompletely resolved.

The role of long-term plasticity in transthalamic pathways is less established. While experience-dependent plasticity refines thalamocortical circuits, both in development[145] and in adults[146–148], experiments isolating the transthalamic subpopulation have not been performed. Whether classical forms of long-term synaptic plasticity operate at the L5 → HO synapse in mature circuits, and how such plasticity would interact with the proposed role of this pathway in reliable signal transmission[149] remains an open question. Addressing these developmental and plastic mechanisms will be essential for understanding how transthalamic pathways contribute to learning and adaptive cortical computation.

### Transthalamic pathways and consciousness

The thalamus has long been implicated in conscious experience[150–153] and transthalamic circuits offer a mechanistic route linking this classical idea with modern theories[154]. Studies in humans underscore that neuronal population dynamics are a requirement for conscious experience[155–157] and

L5 pyramidal neurons have been proposed as crucial integrators[158], supporting earlier theoretical work[159,160]. Their leading framework is the dendritic integration model, in which mGluR activation couples apical and basal dendritic signals on L5 neurons, particularly by HO thalamus[158]. Since feedback transthalamic pathways predominantly activate mGluRs (see Section "Transthalamic pathways: general wiring pattern" and Fig. 1)[24], they may play a critical role in cortical processing underlying consciousness.

Feedforward transthalamic pathways may also contribute to conscious perception. Systems-level theories of consciousness, like the global network workspace theory (GNWT), propose an "ignition" like propagation of neural activity from sensory to more cognitive cortical regions for a given stimulus to reach conscious perception[161–163]. Given that feedforward transthalamic pathways transmit via strong driver synapses, they may be instrumental in sparking sensory-to-cognitive transformations. Note that simultaneous with this "forward" propagation is the activation of feedback transthalamic pathways and, consequently, mGluR activation in the "backward" direction. Together, these features suggest that transthalamic circuits may be central to the integration and propagation processes that make perception conscious.

### Concluding remarks

Over the past decade, advancements in circuit-dissection techniques have dramatically improved our understanding of transthalamic pathways. Since the first functional demonstration in a brain slice[22], circuit technologies in behaving animals has provided compelling evidence that these circuits contribute directly to perception, movement, and cognition. Importantly, evidence is accumulating that cortical layer 5 projections to HO thalamus shape brain-wide computation in ways not achievable through cortico-cortical pathways alone. The notion that entire thalamic nuclei serve uniform functions has been replaced by a more nuanced view, where distinct thalamic circuits contribute differentially to cognition and behavior. A key challenge for the next decade is to directly study the transthalamic pathway as it spans the three brain regions during tasks relevant to predictive processing and perception. This will allow the synthesis of scattered findings across sensory, motor, and cognitive domains into a unified framework. This will require integrating experimental results with theoretical models of cortical computation and consciousness. In doing so, we may find that transthalamic pathways act as flexible hubs that transform, integrate, and gate information across the cortical hierarchy.

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

## Acknowledgements

The authors thank S. Murray Sherman for comments on the manuscript. K.P.K is supported by National Institute of Neurological Disorders and Stroke (Grant NS094184) and C.M is supported by the National Health and Medical Research Council, Australia (Grant GNT2003646). The Florey Institute of Neuroscience and Mental Health acknowledges the strong support from the Victorian Government and in particular the funding from the Operational Infrastructure Support Grant.

## Author contributions

K.P.K.-writing (draft and revisions); C.M.-writing (draft and revisions).

## Competing interests

The authors declare no competing interests.
