## [Transparent Peer Review file · Communications Biology]

The role of transthalamic pathways in perception

Corresponding Author: Dr Christina Mo

Version 1:

Reviewer comments:

Reviewer #1

(Remarks to the Author)

In a cortico-cortical centric milieu, the unique and powerful contributions of cortico-thalamo-cortical pathways deserves attention. For this reason, the review article by Koster and Mo is important. The review includes detailed and critical assessments of recent studies evaluating the representation and causal contributions of various components of the transthalamic pathway.

However, unfortunately, the novelty of this review is not clear. The overlap with another recent review (Sherman and Usrey 2024, Journal of Neuroscience) makes for a hard argument for the necessity of the review under evaluation. The duplication in content is rather extensive, including many key references, underlying anatomical motifs, concepts of efference copy, and GABAergic gating of HO thalamus. The authors acknowledge the Sherman and Usrey review and imply that their review has a larger focus on behavioral relevance vs circuit elements. Nonetheless, the conceptual overlap is widespread.

One feature in particular that I appreciate about the Koster and Mo manuscript is that it addresses head-on the challenges in experimentally isolating the transthalamic pathway, and the corresponding difficulties with interpretations of recent causal studies. This is largely absent from the Sherman and Usrey review.

Other points.

The description of the findings illustrated in Fig. 4 are incomplete. Why would suppression of S1 L5 > P0m reduce stimulus selectivity in S1? It isn't clear to me how this may occur, or how to interpret this in the context of the transthalamic framework.

The content of this sentence is not clear to me, given extensive reciprocal loops in sensory areas (631-632) "While frontal and motor areas appear to more readily form reciprocal connectivity loops, this organization is largely avoided by transthalamic circuits in sensory areas."

I would caution the use of "perception" in the title. Non-human animal studies readily assay (e.g.) detection and discrimination. However, in such studies perception is only implied due to the lack of subjective report. Furthermore, given that some of the processes described are not necessarily perceived (e.g., corollary discharge) it seems like an odd word choice for the title. While the authors do describe implications of transthalamic pathways in theories of consciousness, this is limited to a short section at the end of the review.

Reviewer #2

(Remarks to the Author)

I have been interested in this topic for a long time, so I was looking forward to reading this review paper. To be honest, I was quite disappointed for the following reasons. My understanding is that the transthalamic pathways refer to those through which signals from cortical area A to B are transmitted via the thalamus, instead of the direct pathway from cortical area A to B. The critical question is whether the neurons that receive projections from cortical area A do indeed project to cortical area B. Unless this question is answered, it remains possible that thalamic neurons that receive projections from area A may differ from those that project to area B. This question is important because there is no local excitatory connections within the thalamus. The most studies reviewed in this paper do not clearly answer the question. To unequivocally demonstrate the importance of the transthalamic pathway, one needs to manipulate the disynaptic pathway. Moreover, I did not find any clear answer to the question of whether cortico-cortical and transthalamic pathways carry different information or not. Therefore, to

my disappointment, I did not find reading this long paper valuable.

Version 2:

Reviewer comments:

Reviewer #1

(Remarks to the Author)

Reviewer #3

(Remarks to the Author)

This manuscript reviews cortico-thalamo-cortical (or transthalamic) pathways and their proposed contributions to perceptual processing as defined by performance in behavioral tasks. The authors compile anatomical and physiological evidence of higher-order thalamic relays and provide an organized text that integrates the body of in vivo recording and perturbation studies across sensory, motor and cognitive contexts. A particular strength is the balanced treatment of what current experiments do support about transthalamic involvement in information processing and behavior, alongside a clear discussion of the main interpretational limitations (e.g., incomplete disynaptic pathway isolation, collateral projections, and indirect network effects) and the methodological advances needed to resolve them. Overall, the review provides a useful, experiment-centered framework for readers interested in how the higher-order thalamus may mediate corticocortical communication during cognition. The manuscript is timely and useful, and makes a distinctive contribution relative to Sherman and Usrey (2024) by, first, systematically compiling and explaining experimental evidence and, second, explicitly discussing the limitations and interpretational confounds that currently prevent fully definitive conclusions. This balanced approach is one of the strongest aspects of the manuscript, and makes it valuable even in the presence of that recent review. Below are my remaining comments, mostly aimed at strengthening the positioning and clarity of the claims.

1. Line 88: "In this review, we synthesize these circuit findings (Sherman and Usrey, 2024), with a focus on their behavioral relevance, but acknowledge the methodological limitations. We present emerging evidence that transthalamic pathways are well-positioned as dynamic integrators that transform and broadcast contextual signals, internal states, and task-relevant information across the cortex. By consolidating evidence across circuit physiology, systems neuroscience, and cognitive theory, we provide a conceptual update on thalamic function: from local communication to brain-wide computation (Sherman and Usrey, 2024)."

Here, the authors cite Sherman and Usrey (2024) unnecessarily for a statement that was supported previously in the manuscript by primary sources (in the first sentence), and for another statement that looks like a summary of their review (the last sentence). This matters not only for attribution precision, but also for the perceived redundancy with that review.

2. Line 140: "How HO thalamic neurons implement these circuits remains unresolved. Single-cell tracing shows that many HO neurons are multiareal (branching to multiple cortical areas) (Ohno et al., 2011; Clascá et al., 2012), which suggests that they could participate in both feedforward and feedback pathways simultaneously. Because physiological studies demonstrate that the same nucleus (e.g., POM) provides both driver and modulatory outputs, this would require individual thalamic neurons to target different cortical regions with distinct synaptic properties, which is supported by morphological data (Rodríguez-Moreno et al., 2020) (Figure 2A)."

In this paragraph presenting alternative organizational models, the authors generally explain one of the models, but the text doesn't yet spell out what the evidence actually is (what morphology/tracing/physiology concretely supports the model; please, add primary-source citations. Relatedly, I couldn't find the reference Ohno et al., 2011). I think that adding this information strengthens the position that differentiates this review from Sherman and Usrey, 2024.

3. Line 147: "Alternatively, subpopulations may specialize, such that some HO neurons project exclusively to primary cortex (modulators), others exclusively to higher cortex (drivers), and a third group - the multiareal cells - branch to both (Figure 2B). This mixed organization is most consistent with current data. A purely segregated model, where feedforward and feedback relays are entirely distinct, appears unlikely given the prevalence of multiareal cells (Figure 2C). Clarifying these contributions will be key for understanding how transthalamic signals are temporally and functionally distributed across the cortex."

In the same spirit as Point 2, the authors explain that model B is most consistent with current data; but, no cites are provided. Please, add the papers that support model B and briefly describe the evidence.

4. Line 168: "Transthalamic pathways are disynaptic circuits (the corticothalamic and thalamocortical synapses) that run parallel with other corticothalamic (e.g., those from L6), thalamocortical, and corticocortical circuits. Therefore, selective study of the entire transthalamic pathway is very challenging and to date, has only been achieved in studies using slice electrophysiology and anatomical tracing (Theyel et al., 2010; Mo and Sherman, 2019; Blot et al., 2021; Miller-Hansen and Sherman, 2022; Koster and Sherman, 2024)."

The authors focused mainly on in vivo data supporting, at least partially, the function of the transthalamic pathways in information processing. To introduce these data, the authors provide the paragraph pasted above starting in Line 168. Here the authors explain that the entire pathway has been, indeed, studied using tracings and in vitro ephys approaches, but they did not summarize the results. I think that the reader would benefit from having an explicit explanation of the evidence found in the referenced work.

Reviewer #4

(Remarks to the Author)

This manuscript provides a timely and thoughtful synthesis of recent work examining transthalamic pathways and their contributions to cortical processing and behavior. The authors clearly articulate the conceptual framework surrounding higher-order thalamic function and make a commendable effort to highlight the technical and interpretive limitations that currently constrain the field. In particular, the review does not overstate causal conclusions and explicitly acknowledges the difficulty of isolating strictly transthalamic contributions, which I view as a strength rather than a weakness.

While there is some conceptual overlap with Sherman & Usrey (2024), this is largely unavoidable given shared foundational ideas. Importantly, the present review offers a complementary perspective by emphasizing recent circuit-manipulation studies, behavioral experiments, and the challenges inherent in interpreting these approaches. Overall, the manuscript is well written, informative, and provides a useful synthesis of the current state of the field.

Personally, I would have appreciated a broader treatment of higher-order (HO) thalamic function that extends beyond strictly defined transthalamic pathways. While the authors state that they impose strict boundaries by focusing only on transthalamic routes, several of the studies discussed have not definitively demonstrated that the information being manipulated or recorded is strictly transthalamic in nature. This creates some tension between the stated scope and the literature being evaluated.

Thus, I would have personally preferred a less restrictive organization based on pathway classification (feedforward vs. feedback; transthalamic vs. corticocortical), and instead a more general discussion of emerging evidence for HO thalamic function, including: 1. Correlative imaging and recording studies that reveal activity patterns in HO nuclei during perceptual or cognitive tasks. 2. Circuit-level functions of HO nuclei, including their roles in coordinating cortical areas, shaping temporal dynamics, or modulating gain and synchrony. 3. How these circuit functions might relate to cortical function, beyond the behavioral outcomes in specific sensory detection, discrimination, or sensorimotor tasks that currently show highly variable results.

Such an expanded perspective would have better captured what is currently known about HO thalamus - implicating transthalamic pathways and could help bridge the gap between circuit anatomy, physiology, and perceptual function. These comments are intended primarily as encouragement for future work rather than as required revisions to the present manuscript. It does not take away from the value of the current review, which synthesizes a complex body of literature and clearly articulates existing technical and conceptual limitations.

I have only a few comments that the authors could consider incorporating:

1. Mixed relay model and anatomical evidence (Line 151)

The authors state that a mixed relay model is most consistent with current data but do not cite anterograde or retrograde tracing studies that directly support this idea. It would strengthen this section to reference work using approaches such as CTB or retrograde-AAV labeling (e.g., Bennett et al., *Neuron* 2019; Juavinett et al., *J Comp Neurol* 2020), which provide important anatomical evidence regarding branching patterns and projection targets.

2. Diversity of synapse types in mixed models

In addition to branching patterns, it would be valuable to mention that a single HO neuron projecting to multiple cortical areas may form synapses of different functional classes across those areas (e.g., driver-like vs. modulator-like). Structural evidence from Clascá and functional evidence from Sherman support this idea and would further enrich the concept of "mixed" relay models.

3. Development, experience, and plasticity

The review focuses almost exclusively on the functions of mature transthalamic pathways. However, the authors do not address how these circuits develop, whether their functional roles are shaped by learning or experience, or how HO pathways may participate in cortical plasticity. Even a brief discussion of these issues would broaden the impact of the review and highlight important open questions.

4. Comment on Reviewer 2's Concern (also part of my concern, as stipulated above)

The question of whether corticocortical and transthalamic pathways support distinct functions is indeed important. The authors have added discussion addressing this point, and although the available evidence remains limited, this addition helps to more clearly frame both what is known and what remains unresolved. Given the current state of the field, I think the authors' cautious treatment of this issue is appropriate.

Anthony Holtmaat

Version 3:

Reviewer comments:

Reviewer #3

(Remarks to the Author)

The authors have revised the manuscript carefully and the new version is improved in important respects. In particular, the introduction no longer relies so heavily on Sherman and Usrey (2024), the section on individual HO relay organization now includes clearer anatomical support for the mixed-relay framework, and the section on methodological limitations now provides a useful summary of what has been shown by slice physiology and anatomical reconstruction. I have only one remaining comment.

The authors may wish to cite or add a brief comment on Koster and Sherman (2026, Journal of Neuroscience), which is directly relevant to their section on "Modulation and gating of transthalamic pathways." This recent paper specifically addresses modulation and gating of transthalamic and subcortical pathways through somatosensory thalamus. In my view, the manuscript is ready for publication.

Reviewer #4

(Remarks to the Author)

I appreciate the authors' response to the reviewers' comments. The manuscript provides a helpful overview of the current state of knowledge on transthalamic pathways and synthesizes several compelling conceptual models that will be of interest to the field and may stimulate further experimental work.

Anthony Holtmaat

Reviewers' comments:

Reviewer #1 (Remarks to the Author):

In a cortico-cortical centric milieu, the unique and powerful contributions of cortico-thalamo-cortical pathways deserves attention. For this reason, the review article by Koster and Mo is important. The review includes detailed and critical assessments of recent studies evaluating the representation and causal contributions of various components of the transthalamic pathway.

However, unfortunately, the novelty of this review is not clear. The overlap with another recent review (Sherman and Usrey 2024, Journal of Neuroscience) makes for a hard argument for the necessity of the review under evaluation. The duplication in content is rather extensive, including many key references, underlying anatomical motifs, concepts of efference copy, and GABAergic gating of HO thalamus. The authors acknowledge the Sherman and Usrey review and imply that their review has a larger focus on behavioral relevance vs circuit elements. Nonetheless, the conceptual overlap is widespread.

One feature in particular that I appreciate about the Koster and Mo manuscript is that it addresses head-on the challenges in experimentally isolating the transthalamic pathway, and the corresponding difficulties with interpretations of recent causal studies. This is largely absent from the Sherman and Usrey review.

We thank the reviewer for their careful assessment and for highlighting the aspects of our review that they found particularly valuable. In response to the reviewer's concern about conceptual overlap with Sherman & Usrey (2024), we have reframed some aspects of the review to emphasize distinctness.

The novelty of our review is the focus on recent circuit manipulations and recordings during behavioral tasks, uniquely discussing technical challenges for the future. This was not addressed by Sherman and Usrey (2024) but is key to designing better experiments to understand the function of the pathway and thus moving the field forward.

We focus on the recent increase in functional testing that Usrey and Sherman did not cover in conceptual detail (very brief paragraphs), nor technological detail. Also, the majority of publications that we discuss in our review (64% of Table 1) are key to understanding transthalamic circuitry in cognition but are not cited by Usrey and Sherman: Lam et al., 2024, McKinnon et al., 2025, Moberg et al., 2025, Han & Bonin, 2023, Neske & Cardin, 2023, Takahashi et al., 2022, Alcaez et al., 2018.

We also emphasize that our discussions on transthalamic function are complementary rather than overlapping:

- Usrey and Sherman present a broader discussion on the function of higher-order thalamus in general that indeed includes speculation around GABAergic gating of inputs to HO thalamus. However, in our discussion on this topic, we:

1) present the evidence for other forms of convergence onto HO thalamic neurons and the potential functional consequences, 2) emphasize the gating of cortical L5 inputs to HO thalamus (rather than excitatory inputs in general), and 3) use this discussion of gating as a conduit to discuss a related topic that is not covered in Sherman and Usrey - that tonic inhibition to HO thalamus is a common functional motif that might regulate HO relay firing generally. We now cite the Sherman and Usrey review where overlap is noted (e.g. Figure 6 legend, line 616).

- Usrey and Sherman discuss efference copy, cortical synchrony, the need to classify cortical L5 projection types, the clash with core and matrix principles, and an evolutionary perspective on transthalamic function. In contrast, we discuss stimulus detection/discrimination, maintenance of a percept, external vs self-generated motion and speculate on its role in gating, sensory vs motor vs cognitive functions, and consciousness. The only overlap here is efference copy, which is a major hypothesis of transthalamic function.

As pointed out by Reviewer #1, our review emphasizes the work which uses advanced circuit dissection technologies, paving the way for the very recent functional interpretations. We provide in-depth discussions on these behavioral studies and integrate the current understanding of each sensory/motor/cognitive system, which is perhaps not as progressed as the field had hoped due to the technical difficulties of studying a neural pathway that spans 3 brain regions. Reviewer 2 has emphasized this point in their comments. We believe our review provides a unique resource that highlights the developing state of understanding of this circuit and explicitly discusses the technical and interpretive challenges required to advance this.

Changes

Some overlap in foundational anatomical motifs and key references is unavoidable, as both reviews must be standalone and the same core hypotheses underpin the field.

We now reduce the discussion on gating, referring to more detailed descriptions of the process in other publications. In addition, we have merged figures 6 and 7 into a single illustration. This achieved two things: first, we removed the panel illustrating GABAergic shunting of transthalamic information because it overlapped with Sherman and Usrey. Second, by using the latter part of Figure 6 to directly/visually link GABAergic gating to the disinhibitory motif under discussion, we feel it enhances the complementary emphasis between Sherman and Usrey and our article. We retain a fraction of the features of the original figures because it helps visualise the hypothesized mechanisms. See lines 612-630 (deletions were made) and Figure 6 legend.

We also point out more explicitly that circuit technologies need to be developed along with tasks and analyses so the field can advance in its understanding of the pathway. See highlighted lines 17, 89, 443, 470, 529, 569, and 683 in the manuscript.

The rest of the focus, structure, and what is interpreted differs from that of Sherman & Usrey (2024). We hope the revised manuscript now clearly conveys its distinct value to the field.

Other points.

The description of the findings illustrated in Fig. 4 are incomplete. Why would suppression of S1 L5 > POm reduce stimulus selectivity in S1? It isn't clear to me how this may occur, or how to interpret this in the context of the transthalamic framework.

We thank the reviewer for pointing out that the original explanation was incomplete. Indeed, the figure did not include the POm projection to S1 that is the most likely reason for the effects in S1 (although we cannot rule out that intermediary regions are involved). We have now clarified this logic both in the main text and in the figure/figure legend.

Briefly, suppression of the S1 L5→POm pathway reduces stimulus selectivity in S1 L2/3 likely through the POm→S1 projection (i.e. S1 L5 to POm to S1 L2/3). With reduced driving activity to POm, this may reduce the activity of POm cells which project to S1 L2/3 cells. The relevance to the transthalamic pathway (S1 L5 to POm to S2), is that the transthalamic POm cell is likely the same cell as the one that projects to S1. This is based on single-cell tracing studies showing that all POm cells that project to S2, also projected to S1 (Ohno et al., 2011, discussed in section 2.1). However, although POm to S2 projections are synaptically powerful (driver), POm to S1 projections are weaker (modulator), so their functional effects may differ. This is indeed what we see: S1 L2/3 stimulus response is modestly impacted by inhibition of S1 L5 to POm, compared to substantial loss of selectivity in S2 L2/3 cells (Mo et al. 2024).

We now illustrate this in the figure, describe it in the figure legend and thank the Reviewer for bringing this point to our attention.

Changes highlighted below.

Line 328: To probe this, Mo and colleagues (2024) combined inhibition of S1 L5→POm terminals with two-photon calcium imaging in S1 and S2, thereby recording impacts on the feedforward transthalamic pathway to S2 (as in **Figure 3C**) and also the reciprocal pathway back to S1.

Line 336: In S1, inhibition changed cell discriminability to equal fractions of those selective for the hit and CR textures (**Figure 4B**) (Mo et al., 2024). This more modest impact of inhibiting S1 L5 → POm on S1 responses is expected based on the modulatory synaptic properties of the POm → S1 projection (Viaene et al., 2011b). In

contrast, the substantial loss of selectivity in S2 cells is expected based on the driver properties of the POm → S2 projection (Miller-Hansen & Sherman, 2022).

Figure 4 changed to show feedback modulatory projections to S1.

Figure 4 legend:

B) When the first leg of the somatosensory transthalamic pathway (S1 L5 → POm) is inhibited at the synapse in thalamus, the textures are no longer distinguishable in S1 (likely via S1 L5 → POm → S1) and reversed in S2, aligning with discrimination errors. Adapted from (Mo et al., 2024).

The content of this sentence is not clear to me, given extensive reciprocal loops in sensory areas (631-632) “While frontal and motor areas appear to more readily form reciprocal connectivity loops, this organization is largely avoided by transthalamic circuits in sensory areas.”

The Reviewer is correct. There are many reciprocal loops in sensory areas and the point of emphasis was meant to be that feedforward transthalamic circuits appear less common in frontal and motor areas - this was lost the way the sentence was written. We have now corrected this in the text to reflect the current anatomical data: sensory thalamus shows strong feedforward motifs (e.g., V1→LP→V2, S1→POm→S2), whereas higher-order thalamic nuclei such as MD and VAL participate primarily in recurrent corticothalamic loops.

Line 639: While transthalamic circuits in frontal and motor areas appear to be dominated by reciprocal connectivity loops, sensory transthalamic circuits demonstrate a propensity for forming feedforward and feedback pathways in addition to reciprocal ones

I would caution the use of “perception” in the title. Non-human animal studies readily assay (e.g.) detection and discrimination. However, in such studies perception is only implied due to the lack of subjective report. Furthermore, given that some of the processes described are not necessarily perceived (e.g., corollary discharge) it seems like an odd word choice for the title. While the authors do describe implications of transthalamic pathways in theories of consciousness, this is limited to a short section at the end of the review.

We appreciate the reviewer’s concern about the use of the term “perception” in our title (The role of transthalamic pathways in perception), particularly given the difference between behaviourally measurable detection/discrimination and subjective experience.

In animal work, perception is operationalised through simple stimulus-guided behaviour, as subjective report is unavailable. (We prefer not to enter philosophical

debate of the utility of animal models to understand human perception in this review but we acknowledge the issue raised.) Many processes essential for perception (e.g., efference copy, prediction error signalling) are not consciously experienced yet critically shape perceptual outcomes. The literature also increasingly interprets the contributions of transthalamic pathways within perceptual and predictive processing frameworks.

Subconscious perception and “simple” perception of a stimulus is still perception. We have therefore retained the title while clarifying our operational definition of “perception,” but explicitly note that our focus is on animal studies of perceptual processing rather than subjective phenomenology in humans.

Line 216: Having established the organization of transthalamic pathways and methodological challenges, we examine recent studies in awake rodents that monitor or manipulate these circuits to understand their role in attention, decision making, and sensory perception (Table 1), **which are operationalized by the observable outcomes in goal-directed stimulus-response tasks.**

Reviewer #2 (Remarks to the Author):

I have been interested in this topic for a long time, so I was looking forward to reading this review paper. To be honest, I was quite disappointed for the following reasons. My understanding is that the transthalamic pathways refer to those through which signals from cortical area A to B are transmitted via the thalamus, instead of the direct pathway from cortical area A to B. The critical question is whether the neurons that receive projections from cortical area A do indeed project to cortical area B. Unless this question is answered, it remains possible that thalamic neurons that receive projections from area A may differ from those that project to area B. This question is important because there is no local excitatory connections within the thalamus. The most studies reviewed in this paper do not clearly answer the question. To unequivocally demonstrate the importance of the transthalamic pathway, one needs to manipulate the disynaptic pathway. Moreover, I did not find any clear answer to the question of whether cortico-cortical and transthalamic pathways carry different information or not. Therefore, to my disappointment, I did not find reading this long paper valuable.

We thank the reviewer for articulating what we view as the central unresolved question in this field: whether the neurons receiving corticothalamic input are indeed the same neurons that project to higher-order cortex. We fully agree that unequivocally demonstrating this requires pathway-specific manipulation of both legs of the circuit - something that has not yet been technically achievable and we share the Reviewer’s disappointment in the field’s current understanding of the role of transthalamic pathways. This is the reason why our review is so important right now.

To address this important concern, we dedicated a section to discuss this “Difficulties in isolating the transthalamic pathway for study” (section 2.2), which highlights that:

- existing studies often isolate only one leg of the pathway in experiments
- the limitations of current viral and genetic tools
- the need for next-generation disynaptic targeting strategies

including Figure 3, which depicts ‘Difficulties in experimentally manipulating the transthalamic pathway’.

Regarding a clear answer to the question of whether direct corticocortical and transthalamic pathways carry different information, the answer is yes and we expand our discussion in section 4.4 Comparison of corticocortical and transthalamic pathway function to make this more explicit.

The consensus from these studies is that the corticocortical pathway in the sensory systems are conveying sensory feature information to higher-order cortex, whereas the transthalamic pathway is conveying goal-directed and learned stimulus-response information to be integrated at the level of HO thalamus, to then transmit to HO cortex.

We thank the Reviewer for bringing this need to emphasize this to our attention.

Line 551: Taken together, there is mounting evidence that cortico-subcortical projections from L5 are major determinants of perception, particularly the integration of the rewarded stimulus with action, (Takahashi et al., 2020; Musall et al., 2023; Moberg et al., 2025; Schneider et al., 2025) whilst corticocortical projections to higher-order cortex represent sensory feature information (El-Shamayleh et al., J Neurosci, 2013, Matsui and Ohki, 2013, Glickfield et al., 2013), although stimulus-reward association activity develops with learning (Chen et al., 2015, Chen et al., 2013, Moberg et al., 2025).

Line 569: However, as discussed previously, we await technology that allows manipulation of the complete pathway spanning the three brain regions to make more concrete predictions of transthalamic function. It is important to note that selective silencing of the direct corticocortical pathway is also rare (Chang et al., 2022), which emphasizes the need for more scrupulous dissection of circuitry if we are to accurately assign function.

Response to Reviewers

We thank the two new reviewers for their thoughtful and constructive evaluations of our manuscript. We are particularly grateful for their assessment that the review provides a balanced and necessary synthesis of experimental findings and interpretational limitations in the study of transthalamic pathways, and that it offers a complementary perspective to Sherman & Usrey (2024). We have revised the manuscript accordingly and believe the changes have strengthened its clarity, positioning, and evidentiary grounding.

Reviewers' comments:

Reviewer #3 (Remarks to the Author):

This manuscript reviews cortico-thalamo-cortical (or transthalamic) pathways and their proposed contributions to perceptual processing as defined by performance in behavioral tasks. The authors compile anatomical and physiological evidence of higher-order thalamic relays and provide an organized text that integrates the body of in vivo recording and perturbation studies across sensory, motor and cognitive contexts. A particular strength is the balanced treatment of what current experiments do support about transthalamic involvement in information processing and behavior, alongside a clear discussion of the main interpretational limitations (e.g., incomplete disynaptic pathway isolation, collateral projections, and indirect network effects) and the methodological advances needed to resolve them. Overall, the review provides a useful, experiment-centered framework for readers interested in how the higher-order thalamus may mediate corticocortical communication during cognition. The manuscript is timely and useful, and makes a distinctive contribution relative to Sherman and Usrey (2024) by, first, systematically compiling and explaining experimental evidence and, second, explicitly discussing the limitations and interpretational confounds that currently prevent fully definitive conclusions. This balanced approach is one of the strongest aspects of the manuscript, and makes it valuable even in the presence of that recent review. Below are my remaining comments, mostly aimed at strengthening the positioning and clarity of the claims.

We thank Reviewer #3 for their careful reading and constructive suggestions aimed at strengthening clarity and positioning.

1. Line 88: "In this review, we synthesize these circuit findings (Sherman and Usrey, 2024), with a focus on their behavioral relevance, but acknowledge the methodological limitations. We present emerging evidence that transthalamic pathways are well-positioned as dynamic integrators that transform and broadcast contextual signals, internal states, and task-relevant information across the cortex. By consolidating evidence across circuit physiology, systems neuroscience, and cognitive theory, we provide a conceptual update on thalamic function: from local communication to brain-wide computation (Sherman and Usrey, 2024)."

Here, the authors cite Sherman and Usrey (2024) unnecessarily for a statement that was supported previously in the manuscript by primary sources (in the first sentence), and for another statement that looks like a summary of their review (the last sentence). This matters not only for attribution precision, but also for the perceived redundancy

with

that

review.

We appreciate this point regarding attribution precision and perceived redundancy. We have revised this paragraph (line 83) to remove unnecessary citations to Sherman & Usrey (2024), particularly where primary sources had already been cited or where the conceptual framing reflects our synthesis rather than a summary of their review. This change clarifies the distinction between the two reviews and avoids over-attribution.

2. Line 140: “How HO thalamic neurons implement these circuits remains unresolved. Single-cell tracing shows that many HO neurons are multiareal (branching to multiple cortical areas) (Ohno et al., 2011; Clascá et al., 2012), which suggests that they could participate in both feedforward and feedback pathways simultaneously. Because physiological studies demonstrate that the same nucleus (e.g., POm) provides both driver and modulatory outputs, this would require individual thalamic neurons to target different cortical regions with distinct synaptic properties, which is supported by morphological data (Rodriguez-Moreno et al., 2020) (Figure 2A).”

In this paragraph presenting alternative organizational models, the authors generally explain one of the models, but the text doesn't yet spell out what the evidence actually is (what morphology/tracing/physiology concretely supports the model; please, add primary-source citations. Relatedly, I couldn't find the reference Ohno et al., 2011). I think that adding this information strengthens the position that differentiates this review from Sherman and Usrey, 2024.

We thank the reviewer for noting that this section (line 129) would benefit from more explicit description of the supporting evidence. We have:

- Corrected the citation (Ohno et al., *Cerebral Cortex*, 2012).
- Added brief descriptions of the methodologies used in Ohno et al. (2012) and Rodriguez-Moreno et al. (2020), including viral-based single-cell tracing and in vivo electroporation approaches used to reconstruct projection patterns of individual POm neurons.
- Clarified how these morphological findings support the possibility of multiareal branching and differential targeting.

These additions strengthen the evidentiary basis of the organizational models discussed.

3. Line 147: “Alternatively, subpopulations may specialize, such that some HO neurons project exclusively to primary cortex (modulators), others exclusively to higher cortex (drivers), and a third group - the multiareal cells - branch to both (Figure 2B). This mixed organization is most consistent with current data. A purely segregated model, where feedforward and feedback relays are entirely distinct, appears unlikely given the prevalence of multiareal cells (Figure 2C). Clarifying these contributions will be key for understanding how transthalamic signals are temporally and functionally distributed across the cortex.”

In the same spirit as Point 2, the authors explain that model B is most consistent with current data; but, no cites are provided. Please, add the papers that support model B and briefly describe the evidence.

We agree that the statement that the mixed model is most consistent with current data (line 143 onwards) should be explicitly supported. We have now added citations and a brief description of the anatomical and single-cell tracing evidence indicating the prevalence of multiareal higher-order neurons, which argues against a strictly segregated relay model. This revision clarifies the empirical grounding of our interpretation.

4. Line 168: “Transthalamic pathways are disynaptic circuits (the corticothalamic and thalamocortical synapses) that run parallel with other corticothalamic (e.g., those from L6), thalamocortical, and corticocortical circuits. Therefore, selective study of the entire transthalamic pathway is very challenging and to date, has only been achieved in studies using slice electrophysiology and anatomical tracing (Theyel et al., 2010; Mo and Sherman, 2019; Blot et al., 2021; Miller-Hansen and Sherman, 2022; Koster and Sherman, 2024).”

The authors focused mainly on in vivo data supporting, at least partially, the function of the transthalamic pathways in information processing. To introduce these data, the authors provide the paragraph pasted above starting in Line 168. Here the authors explain that the entire pathway has been, indeed, studied using tracings and in vitro ephys approaches, but they did not summarize the results. I think that the reader would benefit from having an explicit explanation of the evidence found in the referenced work.

We appreciate this suggestion. While these studies are discussed in more detail later in the review, we agree that readers would benefit from a brief summary at this earlier point. We have therefore added (line 174) a concise description of the key findings from Theyel et al. (2010), Mo & Sherman (2019), Miller-Hansen (2022), and related studies, explicitly stating what has been demonstrated regarding disynaptic cortico–thalamo–cortical transmission in vitro and through anatomical reconstruction. This provides clearer context before transitioning to in vivo limitations.

Reviewer #4 (Remarks to the Author):

This manuscript provides a timely and thoughtful synthesis of recent work examining transthalamic pathways and their contributions to cortical processing and behavior. The authors clearly articulate the conceptual framework surrounding higher-order thalamic function and make a commendable effort to highlight the technical and interpretive limitations that currently constrain the field. In particular, the review does not overstate causal conclusions and explicitly acknowledges the difficulty of isolating strictly transthalamic contributions, which I view as a strength rather than a weakness.

While there is some conceptual overlap with Sherman & Usrey (2024), this is largely unavoidable given shared foundational ideas. Importantly, the present review offers a complementary perspective by emphasizing recent circuit-manipulation studies, behavioral experiments, and the challenges inherent in interpreting these approaches. Overall, the manuscript is well written, informative, and provides a useful synthesis of the current state of the field.

We thank Reviewer #4 for their generous and thoughtful assessment of the manuscript and for recognizing the value of our cautious and balanced approach.

Personally, I would have appreciated a broader treatment of higher-order (HO) thalamic function that extends beyond strictly defined transthalamic pathways. While the authors state that they impose strict boundaries by focusing only on transthalamic routes, several of the studies discussed have not definitively demonstrated that the information being manipulated or recorded is strictly transthalamic in nature. This creates some tension between the stated scope and the literature being evaluated.

Thus, I would have personally preferred a less restrictive organization based on pathway classification (feedforward vs. feedback; transthalamic vs. corticocortical), and instead a more general discussion of emerging evidence for HO thalamic function, including: 1. Correlative imaging and recording studies that reveal activity patterns in HO nuclei during perceptual or cognitive tasks. 2. Circuit-level functions of HO nuclei, including their roles in coordinating cortical areas, shaping temporal dynamics, or modulating gain and synchrony. 3. How these circuit functions might relate to cortical function, beyond the behavioral outcomes in specific sensory detection, discrimination, or sensorimotor tasks that currently show highly variable results.

Such an expanded perspective would have better captured what is currently known about HO thalamus - implicating transthalamic pathways and could help bridge the gap between circuit anatomy, physiology, and perceptual function. These comments are intended primarily as encouragement for future work rather than as required revisions to the present manuscript. It does not take away from the value of the current review, which synthesizes a complex body of literature and clearly articulates existing technical and conceptual limitations.

We appreciate the reviewer's perspective regarding a broader treatment of higher-order thalamic function. Our decision to maintain a pathway-based organizational structure was intentional, as the central goal of the review is to evaluate what can be inferred specifically about transthalamic circuits. We agree that some of the studies discussed do not definitively isolate strictly transthalamic contributions; indeed, this methodological limitation is one of the central themes of the review.

To clarify this tension, we acknowledged that many experimental approaches cannot isolate exclusively transthalamic signaling, and that interpretations therefore often reflect broader higher-order thalamic contributions. We believe this strengthens conceptual clarity while maintaining the focused scope of the review.

I have only a few comments that the authors could consider incorporating: 1. Mixed relay model and anatomical evidence (Line 151). The authors state that a mixed relay model is most consistent with current data but do not cite anterograde or retrograde tracing studies that directly support this idea. It would strengthen this section to reference work using approaches such as CTB or retrograde-AAV labeling (e.g., Bennett et al., *Neuron* 2019; Juavinett et al., *J Comp Neurol* 2020), which provide important anatomical evidence regarding branching patterns and projection targets.

We thank the reviewer for these suggested references. We have added a citation (line

150) to the study by Juavinett and colleagues (Bennett et al., is cited elsewhere but we feel it did not contribute to the argument in that section) while clarifying that single-cell reconstructions provide the strongest evidence for branching patterns that inform the mixed relay model. We agree that, taken together, the data more roundly support the mixed-relay model and this is reflected in the text.

2. Diversity of synapse types in mixed models. In addition to branching patterns, it would be valuable to mention that a single HO neuron projecting to multiple cortical areas may form synapses of different functional classes across those areas (e.g., driver-like vs. modulator-like). Structural evidence from Clascá and functional evidence from Sherman support this idea and would further enrich the concept of “mixed” relay models.

We agree this is an important point. We have revised the text (line135) to more clearly state that multiareal higher-order neurons may form synapses with distinct functional properties across cortical targets, referencing structural evidence (e.g., Clascá) and functional classifications (e.g., Sherman) that support this possibility. This clarification strengthens the conceptual framing of the mixed model.

3. Development, experience, and plasticity. The review focuses almost exclusively on the functions of mature transthalamic pathways. However, the authors do not address how these circuits develop, whether their functional roles are shaped by learning or experience, or how HO pathways may participate in cortical plasticity. Even a brief discussion of these issues would broaden the impact of the review and highlight important open questions.

We thank the reviewer for highlighting this important area. While a comprehensive treatment of development and plasticity is beyond the scope of the current review, we have added a brief section in the Future Perspectives section (line 608 ‘Development and plasticity of transthalamic pathways), noting that the development, experience-dependence, and plasticity of transthalamic pathways represent important open questions. This addition broadens the perspective of the review and highlights future directions.

4. Comment on Reviewer 2’s Concern (also part of my concern, as stipulated above) The question of whether corticocortical and transthalamic pathways support distinct functions is indeed important. The authors have added discussion addressing this point, and although the available evidence remains limited, this addition helps to more clearly frame both what is known and what remains unresolved. Given the current state of the field, I think the authors’ cautious treatment of this issue is appropriate.

We appreciate the reviewer’s positive assessment of our cautious framing of this issue. We have retained this balanced discussion and slightly clarified the wording to emphasize the current limitations of available evidence.

REVIEWERS' COMMENTS:

Reviewer #3 (Remarks to the Author):

The authors have revised the manuscript carefully and the new version is improved in important respects. In particular, the introduction no longer relies so heavily on Sherman and Usrey (2024), the section on individual HO relay organization now includes clearer anatomical support for the mixed-relay framework, and the section on methodological limitations now provides a useful summary of what has been shown by slice physiology and anatomical reconstruction. I have only one remaining comment. The authors may wish to cite or add a brief comment on Koster and Sherman (2026, *Journal of Neuroscience*), which is directly relevant to their section on “Modulation and gating of transthalamic pathways.” This recent paper specifically addresses modulation and gating of transthalamic and subcortical pathways through somatosensory thalamus.

In my view, the manuscript is ready for publication.

The reviewer makes a good point and the requested brief comment and citation has been added to the final version (lines 476-477). We thank the reviewer for their time and useful comments throughout this process.

Reviewer #4 (Remarks to the Author):

I appreciate the authors' response to the reviewers' comments. The manuscript provides a helpful overview of the current state of knowledge on transthalamic pathways and synthesizes several compelling conceptual models that will be of interest to the field and may stimulate further experimental work.

We thank the reviewer for their time and useful comments throughout this process.